# Federated Learning of a Mixture of Global and Local Models

## Abstract

We propose a new optimization formulation for training federated learning models. The standard formulation has the form of an empirical risk minimization problem constructed to find a single global model trained from the private data stored across all participating devices. In contrast, our formulation seeks an explicit trade-off between this traditional global model and the local models, which can be learned by each device from its own private data without any communication. Further, we develop several efficient variants of SGD (with and without partial participation and with and without variance reduction) for solving the new formulation and prove communication complexity guarantees. Notably, our methods are similar but not identical to federated averaging / local SGD, thus shedding some light on the essence of the elusive method. In particular, our methods do not perform full averaging steps and instead merely take steps towards averaging. We argue for the benefits of this new paradigm for federated learning.

## 1 Introduction

With the proliferation of mobile phones, wearable devices, tablets, and smart home devices comes an increase in the volume of data captured and stored on them. This data contains a wealth of potentially useful information to the owners of these devices, and more so if appropriate machine learning models could be trained on the heterogeneous data stored across the network of such devices. The traditional approach involves moving the relevant data to a data center where centralized machine learning techniques can be efficiently applied (Dean et al., 2012; Reddi et al., 2016). However, this approach is not without issues. First, many device users are increasingly sensitive to privacy concerns and prefer their data to never leave their devices. Second, moving data from their place of origin to a centralized location is very inefficient in terms of energy and time.

### 1.1 Federated learning

*Federated learning (FL)* (McMahan et al., 2016; Konečný et al., 2016b;a; McMahan et al., 2017) has emerged as an interdisciplinary field focused on addressing these issues by training machine learning models directly on edge devices. The currently prevalent paradigm (Li et al., 2019; Kairouz et al., 2019) casts supervised FL as an empirical risk minimization problem of the form

$$\min_{x \in \mathbb{R}^d} \frac{1}{n} \sum_{i=1}^{n} f_i(x), \tag{1}$$

where $n$ is the number of devices participating in training, $x \in \mathbb{R}^d$ encodes the $d$ parameters of a *global model* (e.g., weights of a neural network) and

$$f_i(x) := \mathbb{E}_{\xi \sim \mathcal{D}_i} [f(x, \xi)]$$

represents the aggregate loss of model $x$ on the local data represented by distribution $\mathcal{D}_i$ stored on device $i$. One of the defining characteristics of FL is that the data distributions $\mathcal{D}_i$ may possess very different properties across the devices. Hence, any potential FL method is explicitly required to be able to work under the *heterogeneous* data setting.

The most popular method for solving (1) in the context of FL is the FedAvg algorithm (McMahan et al., 2016). In its most simple form, when one does not employ partial participation, model compression, or stochastic approximation, FedAvg reduces to Local Gradient Descent (LGD) (Khaled

et al., 2019; 2020), which is an extension of GD performing more than a single gradient step on each device before aggregation. FedAvg has been shown to work well empirically, particularly for non-convex problems, but comes with poor convergence guarantees compared to the non-local counterparts when data are heterogeneous.

**Some issues with current approaches to FL**

The first motivation for our research comes from the appreciation that data heterogeneity does not merely present challenges to the design of new provably efficient training methods for solving (1), but *also inevitably raises questions about the utility of such a global solution to individual users.* Indeed, a global model trained across all the data from all devices might be so removed from the typical data and usage patterns experienced by an individual user as to render it virtually useless. This issue has been observed before, and various approaches have been proposed to address it. For instance, the MOCHA (Smith et al., 2017) framework uses a multi-task learning approach to allow for personalization. Next, (Khodak et al., 2019) propose a generic online algorithm for gradient-based parameter-transfer meta-learning and demonstrate improved practical performance over FedAvg (McMahan et al., 2017). Approaches based on variational inference (Corinzia & Buhmann, 2019), cyclic patterns in practical FL data sampling (Eichner et al., 2019) transfer learning (Zhao et al., 2018) and explicit model mixing (Peterson et al., 2019) have been proposed.

The second motivation for our work is the realization that even very simple variants of FedAvg, such as LGD, which should be easier to analyze, fail to provide theoretical improvements in communication complexity over their non-local cousins, in this case, GD (Khaled et al., 2019; 2020).[1]

This observation is at odds with the practical success of local methods in FL. This leads us to ask the question: *if LGD does not theoretically improve upon GD as a solver for the traditional global problem* (1)*, perhaps LGD should not be seen as a method for solving* (1) *at all. In such a case, what problem does LGD solve?* A good answer to this question would shed light on the workings of LGD, and by analogy, on the role local steps play in more elaborate FL methods such as local SGD (Stich, 2020; Khaled et al., 2020) and FedAvg.

## 2 CONTRIBUTIONS

In our work we argue that the two motivations mentioned in the introduction point in the same direction, i.e., we show that *a single solution can be devised addressing both problems at the same time.* Our main contributions are:

⋄ **New formulation of FL which seeks an implicit mixture of global and local models.** We propose a *new optimization formulation of FL.* Instead of learning a single global model by solving (1), we propose to learn a mixture of the global model and the purely local models which can be trained by each device $i$ using its data $\mathcal{D}_i$ only. Our formulation (see Sec. 3) lifts the problem from $\mathbb{R}^d$ to $\mathbb{R}^{nd}$, allowing each device $i$ to learn a personalized model $x_i \in \mathbb{R}^d$. These personalized models are encouraged to not depart too much from their mean by the inclusion of a quadratic penalty $\psi$ multiplied by a parameter $\lambda \geq 0$. Admittedly, the idea of softly-enforced similarity of the local models was already introduced in the domain of the multi-task relationship learning (Zhang & Yeung, 2010; Liu et al., 2017; Wang et al., 2018) and distributed optimization (Lan et al., 2018; Gorbunov et al., 2019; Zhang et al., 2015). The mixture objective we propose (see (2)) is a special case of their setup, which justifies our approach from the modeling perspective. Note that Zhang et al. (2015); Liu et al. (2017); Wang et al. (2018) provide efficient algorithms to solve the mixture objective already. However, none of the mentioned papers consider the FL application, nor they shed a light on the communication complexity of LGD algorithms, which we do in our work.

⋄ **Theoretical properties of the new formulation.** We study the properties of the optimal solution of our formulation, thus developing an algorithmic-free theory. When the penalty parameter is set to zero, then obviously, each device is allowed to train their own model without any dependence on the data stored on other devices. Such purely local models are rarely useful. We prove that the optimal

---

[1]After our paper was completed, a lower bound on the performance of local SGD was presented that is worse than the known minibatch SGD guarantee (Woodworth et al., 2020a), confirming that the local methods do not outperform their non-local counterparts in the heterogeneous setup. Similarly, the benefit of local methods in the non-heterogeneous scenario was questioned in (Woodworth et al., 2020b).

local models converge to the traditional global model characterized by (1) at the rate $\mathcal{O}(1/\lambda)$. We also show that the total loss evaluated at the local models is never higher than the total loss evaluated at the global model (see Thm. 3.1). Moreover, we prove an insightful structural result for the optimal local models: the optimal model learned by device $i$ arises by subtracting the gradient of the loss function stored on that device evaluated at the same point (i.e., a local model) from the average of the optimal local models (see Thm. 3.2). As a byproduct, this theoretical result sheds new light on the key update step in the model agnostic meta-learning (MAML) method (Finn et al., 2017), which has a similar but subtly different structure.[2] The subtle difference is that the MAML update obtains the local model by subtracting the gradient evaluated at the *global* model. While MAML was originally proposed as a heuristic, we provide rigorous theoretical guarantees.

$\diamond$ **Loopless LGD: non-uniform SGD applied to our formulation.** We then propose a randomized gradient-based method—*Loopless Local Gradient Descent (L2GD)*—for solving our new formulation (Algorithm 1). This method is, in fact, a non-standard application of SGD to our problem, and can be seen as an instance of SGD with non-uniform sampling applied to the problem of minimizing the sum of two convex functions (Zhao & Zhang, 2015; Gower et al., 2019): the average loss, and the penalty. When the loss function is selected by the randomness in our SGD method, the stochastic gradient step can be *interpreted* as the execution of a single local GD step on each device. Since we set the probability of the loss being sampled to be high, this step is typically repeated multiple times, resulting in multiple local GD steps. In contrast to standard LGD, the number of local steps is not fixed, but random, and follows a geometric distribution. This mechanism is similar in spirit to how the recently proposed loopless variants of SVRG (Hofmann et al., 2015; Kovalev et al., 2020) work in comparison with the original SVRG (Johnson & Zhang, 2013a; Xiao & Zhang, 2014). Once the penalty is sampled by our method, the resultant SGD step can be interpreted as the execution of an aggregation step. In contrast with standard aggregation, which performs full averaging of the local models, our method merely takes a *step towards averaging*. However, the step is relatively large.

$\diamond$ **Convergence theory.** By adapting the general theory from (Gower et al., 2019) to our setting, we obtain theoretical convergence guarantees assuming that each $f_i$ is $L$-smooth and $\mu$-strongly convex (see Thm. 4.2). Interestingly, by optimizing the probability of sampling the penalty (we get $p^\star = \frac{\lambda}{\lambda + L}$), which is an indirect way of fixing the *expected number of local steps* to $1 + \frac{L}{\lambda}$, we prove an $\frac{2\lambda}{\lambda + L} \frac{L}{\mu} \log \frac{1}{\varepsilon}$ bound on the expected number of communication rounds (see Cor. 4.3). We believe that this is remarkable in several ways. By choosing $\lambda$ small, we tilt our goal towards pure local models: the number of communication rounds is tending to 0 as $\lambda \to 0$. If $\lambda \to \infty$, the solution our formulation converges to is the optimal global model, and L2GD obtains the communication bound $\mathcal{O}\left( \frac{L}{\mu} \log \frac{1}{\varepsilon} \right)$, which matches the efficiency of GD.

$\diamond$ **What problem do local methods solve?** Noting that L2GD is a (mildly nonstandard) version of LGD,[3] which is a key method most local methods for FL are based on, and noting that, as we show, L2GD solves our new formulation of FL, we offer a new and surprising interpretation of the role of local steps in FL. In particular, the role of local steps in gradient type methods, such as GD, is not to reduce communication complexity, as is generally believed. Indeed, there is no theoretical result supporting this claim in the key heterogeneous data regime. Instead, their role is to steer the method towards finding a mixture of the traditional global and the purely local models. Given that the stepsize is fixed, the more local steps are taken, the more we bias the method towards the purely local models. Our new optimization formulation of FL formalizes this as it defines the problem that local methods, in this case L2GD, solve. There is an added benefit here: the more we want our formulation to be biased towards purely local models (i.e., the smaller the penalty parameter $\lambda$ is), the more local steps does L2GD take, and the better the total communication complexity of L2GD becomes. Hence, despite a lot of research on this topic, our paper provides *the first proof that a local method (e.g., L2GD) can be better than its non-local counterpart (e.g., GD) in terms of total communication complexity in the heterogeneous data setting.* We are able to do this by noting that local methods should better be seen as methods for solving the new FL formulation proposed here.

$\diamond$ **Generalizations: partial participation, local SGD and variance reduction.** We further generalize and improve our method by allowing for (i) stochastic *partial participation* of devices in each communication round,(ii) *subsampling* on each device which means we can perform local SGD steps

---

[2]The connection of FL and multi-task meta learning is discussed in (Kairouz et al., 2019), for example.

[3]To be specific, L2GD is equivalent to Overlap LGD (Wang et al., 2020) with random local loop size.

instead of local GD steps, and (iii) *total variance reduction mechanism* to tackle the variance coming from three sources: locality of the updates induced by non-uniform sampling (already present in L2GD), partial participation and local subsampling. Due to its level of generality, this method, which we call L2SGD++, is presented in the Appendix only, alongside the associated complexity results. In the main body of this paper, we instead present a simplified version thereof, which we call L2SGD+ (Algorithm 3). The convergence theory for it is presented in Thm. 5.1 and Cor. 5.2.

◇ **Heterogeneous data.** All our methods and convergence results allow for fully heterogeneous data and do not depend on any assumptions on data similarity across the devices.

◇ **Superior empirical performance.** We show through ample numerical experiments that our theoretical predictions can be observed in practice.

## 3 NEW FORMULATION OF FL

We now introduce our new formulation for training supervised FL models:

$$
\min_{x_1,\ldots,x_n\in\mathbb{R}^d}\{F(x):=f(x)+\lambda\psi(x)\}
$$

$$
f(x):=\tfrac{1}{n}\sum_{i=1}^n f_i(x_i),\quad \psi(x):=\tfrac{1}{2n}\sum_{i=1}^n\|x_i-\bar{x}\|^2, \tag{2}
$$

where $\lambda\geq 0$ is a penalty parameter, $x_1,\ldots,x_n\in\mathbb{R}^d$ are local models, $x:=(x_1,x_2,\ldots,x_n)\in\mathbb{R}^{nd}$ and $\bar{x}:=\tfrac{1}{n}\sum_{i=1}^n x_i$ is the average of the local models.

Due to the assumptions on $f_i$ we will make in Sec. 3.1, $F$ is strongly convex and hence (2) has a unique solution, which we denote $x(\lambda):=(x_1(\lambda),\ldots,x_n(\lambda))\in\mathbb{R}^{nd}$. We further let $\bar{x}(\lambda):=\tfrac{1}{n}\sum_{i=1}^n x_i(\lambda)$. We now comment on the rationale behind the new formulation.

**Local models ($\lambda=0$).** Note that for each $i$, $x_i(0)$ solves the *local problem* $\min_{x_i\in\mathbb{R}^d} f_i(x_i)$. That is, $x_i(0)$ is the local model based on data $\mathcal{D}_i$ stored on device $i$ only. This model can be computed by device $i$ without any communication whatsoever. Typically, $\mathcal{D}_i$ is not rich enough for this local model to be useful. In order to learn a better model, one has to take into account the date from other clients as well. This, however, requires communication.

**Mixed models ($\lambda\in(0,\infty)$).** As $\lambda$ increases, the penalty $\lambda\psi(x)$ has an increasingly more substantial effect, and communication is needed to ensure that the models are not too dissimilar, as otherwise the penalty $\lambda\psi(x)$ would be too large.

**Global model ($\lambda=\infty$).** Let us now look at the limit case $\lambda\to\infty$. Intuitively, this limit case should force the optimal local models to be mutually identical, while minimizing the loss $f$. In particular, this limit case will solve[4] $\min\left\{f(x)\ :\ x_1,\ldots,x_n\in\mathbb{R}^d,\ x_1=x_2=\cdots=x_n\right\}$, which is equivalent to the global formulation (2). Because of this, let us define $x_i(\infty)$ for each $i$ to be the optimal global solution of (1), and let $x(\infty):=(x_1(\infty),\ldots,x_n(\infty))$.

### 3.1 TECHNICAL PRELIMINARIES

We make the following assumption on the functions $f_i$:

**Assumption 3.1** *For each $i$, the function $f_i:\mathbb{R}^d\to\mathbb{R}$ is $L$-smooth and $\mu$-strongly convex.*

For $x_i,y_i\in\mathbb{R}^d$, $\langle x_i,y_i\rangle$ denotes the standard inner product and $\|x\|:=\langle x_i,x_i\rangle^{1/2}$ is the standard Euclidean norm. For vectors $x=(x_1,\ldots,x_n)\in\mathbb{R}^{nd}$, $y=(y_1,\ldots,y_n)\in\mathbb{R}^{nd}$ we define the standard inner product and norm via $\langle x,y\rangle:=\sum_{i=1}^n\langle x_i,y_i\rangle, \|x\|^2:=\sum_{i=1}^n\|x_i\|^2$. Note that the separable structure of $f$ implies that $(\nabla f(x))_i=\tfrac{1}{n}\nabla f_i(x_i)$, i.e., $\nabla f(x)=\tfrac{1}{n}(\nabla f_1(x_1),\nabla f_2(x_2),\ldots,\nabla f_n(x_n))$.

Note that Assumption 3.1 implies that $f$ is $L_f$-smooth with $L_f:=\tfrac{L}{n}$ and $\mu_f$-strongly convex with $\mu_f:=\tfrac{\mu}{n}$. Clearly, $\psi$ is convex by construction. It can be shown that $\psi$ is $L_\psi$-smooth with $L_\psi=\tfrac{1}{n}$

---

[4]If $\lambda=\infty$ and $x_1=x_2=\cdots=x_n$ does not hold, we have $F(x)=\infty$. Therefore, we can restrict ourselves on set $x_1=x_2=\cdots=x_n$ without loss of generality.

(see Appendix). We can also easily see that $(\nabla\psi(x))_i = \frac{1}{n}(x_i - \bar{x})$(see Appendix), which implies $\psi(x) = \frac{n}{2} \sum_{i=1}^{n} \|(\nabla\psi(x))_i\|^2 = \frac{n}{2} \|\nabla\psi(x)\|^2$.

### 3.2 CHARACTERIZATION OF OPTIMAL SOLUTIONS

Our first result describes the behavior of $f(x(\lambda))$ and $\psi(x(\lambda))$ as a function of $\lambda$.

**Theorem 3.1** *The function $\lambda \to \psi(x(\lambda))$ is non-increasing, and for all $\lambda > 0$ we have*

$$\psi(x(\lambda)) \leq \frac{f(x(\infty)) - f(x(0))}{\lambda}. \tag{3}$$

*Moreover, the function $\lambda \to f(x(\lambda))$ is non-decreasing, and for all $\lambda \geq 0$ we have*

$$f(x(\lambda)) \leq f(x(\infty)). \tag{4}$$

Ineq. (3) says that the penalty decreases to zero as $\lambda$ grows, and hence the optimal local models $x_i(\lambda)$ are increasingly similar as $\lambda$ grows. The second statement suggest that the loss $f(x(\lambda))$ increases with $\lambda$, but never exceeds the optimal global loss $f(x(\infty))$ of the standard FL formulation (1).

We now characterize the optimal local models which connect our model to the MAML framework (Finn et al., 2017), as mentioned in the introduction.

**Theorem 3.2** *For each $\lambda > 0$ and $1 \leq i \leq n$ we have*

$$x_i(\lambda) = \bar{x}(\lambda) - \frac{1}{\lambda}\nabla f_i(x_i(\lambda)). \tag{5}$$

*Further, we have $\sum_{i=1}^{n} \nabla f_i(x_i(\lambda)) = 0$ and $\psi(x(\lambda)) = \frac{1}{2\lambda^2} \|\nabla f(x(\lambda))\|^2$.*

The optimal local models (5) are obtained from the average model by subtracting a multiple of the local gradient. Observe that the local gradients always sum up to zero at optimality. This is obviously true for $\lambda = \infty$, but it is a bit less obvious that this holds for any $\lambda > 0$.

Next, we argue the optimal local models converge to the traditional FL solution at the rate $\mathcal{O}(1/\lambda)$.

**Theorem 3.3** *Let $P(z) := \frac{1}{n} \sum_{i=1}^{n} f_i(z)$. Then, $x(\infty)$ is the unique minimizer of $P$ and we have*

$$\|\nabla P(\bar{x}(\lambda))\|^2 \leq \frac{2L^2(f(x(\infty)) - f(x(0)))}{\lambda}. \tag{6}$$

## 4   L2GD: LOOPLESS LOCAL GD

In this section we describe a new randomized method for solving the formulation (2). Our method is a non-uniform SGD for (2) seen as a 2-sum problem, sampling either $\nabla f$ or $\nabla\psi$ to estimate $\nabla F$. Letting $0 < p < 1$, we define a stochastic gradient of $F$ at $x \in \mathbb{R}^{nd}$ as follows

$$G(x) := \begin{cases} \frac{\nabla f(x)}{1-p} & \text{with probability} \quad 1-p \\ \frac{\lambda\nabla\psi(x)}{p} & \text{with probability} \quad p \end{cases}. \tag{7}$$

Clearly, $G(x)$ is an unbiased estimator of $\nabla F(x)$. This leads to the following method for minimizing $F$, which we call L2GD: $x^{k+1} = x^k - \alpha G(x^k)$. Plugging the formulas for $\nabla f(x)$ and $\nabla\psi(x)$ into (7), and writing the resulting method in a distributed manner, we arrive at Algorithm 1. In each iteration, a coin $\xi$ is tossed and lands 1 with probability $p$ and 0 with probability $1 - p$. If $\xi = 0$, all Devices perform one local GD step (8), and if $\xi = 1$, Master shifts each local model towards the average via (9). As we shall see in Sec. 4.2, our theory limits the value of the stepsize $\alpha$, which has the effect that the ratio $\frac{\alpha\lambda}{np}$ cannot exceed $\frac{1}{2}$. Hence, (9) is a convex combination of $x_i^k$ and $\bar{x}^k$.

Note that Algorithm 1 is only required to communicate when a two consecutive coin tosses land a different value (see the detailed explanation in Sec. C.1 of the appendix). Consequently, the expected number of communication rounds in $k$ iterations of L2GD is $p(1 - p)k$.

**Remark 4.1** *Our algorithm statements do not take the data privacy into the consideration. While privacy is a very important aspect of FL; in this paper, we tackle different FL challenges and thus we ignore privacy issues. However, the proposed algorithms can be implemented in a private fashion as well using tricks that are used in the classical FL scenario (Bonawitz et al., 2017).*

---

**Algorithm 1** L2GD: Loopless Local Gradient Descent

---

**Input:** $x_1^0 = \cdots = x_n^0 \in \mathbb{R}^d$, stepsize $\alpha$, probability $p$
**for** $k = 0, 1, \ldots$ **do**
   $\xi = 1$ with probability $p$ and 0 with probability $1 - p$
   **if** $\xi = 0$ **then**
     All Devices $i = 1, \ldots, n$ perform a local GD step:
$$x_i^{k+1} = x_i^k - \frac{\alpha}{n(1-p)} \nabla f_i(x_i^k) \tag{8}$$
   **else**
     Master computes the average $\bar{x}^k = \frac{1}{n} \sum\limits_{i=1}^{n} x_i^k$
     Master for each $i$ computes step towards aggregation
$$x_i^{k+1} = \left(1 - \frac{\alpha\lambda}{np}\right) x_i^k + \frac{\alpha\lambda}{np} \bar{x}^k \tag{9}$$
   **end if**
**end for**

---

### 4.1 The dynamics of local GD and averaging steps

Notice that *the average of the local models does not change* during an aggregation step. Indeed, $\bar{x}^{k+1}$ is equal to $\frac{1}{n} \sum_{i=1}^{n} x_i^{k+1} \overset{(9)}{=} \frac{1}{n} \sum_{i=1}^{n} \left[\left(1 - \frac{\alpha\lambda}{np}\right) x_i^k + \frac{\alpha\lambda}{np} \bar{x}^k\right] = \bar{x}^k$.

If several averaging steps take place in a sequence, the point $a = \bar{x}^k$ in (9) remains unchanged, and each local model $x_i^k$ merely moves along the line joining the initial value of the local model at the start of the sequence and $a$, with each step pushing $x_i^k$ closer to the average $a$.

*In summary, the more local GD steps are taken, the closer the local models get to the pure local models; and the more averaging steps are taken, the closer the local models get to their average value. The relative number of local GD vs. averaging steps is controlled by the parameter $p$: the expected # of local GD steps is $\frac{1}{p}$, and the expected number of consecutive aggregation steps is $\frac{1}{1-p}$.*

### 4.2 Convergence theory

We now present our convergence result for L2GD.

**Theorem 4.2** *Let Assumption 3.1 hold. If $\alpha \leq \frac{1}{2\mathcal{L}}$, then*
$$\mathbb{E}\left[\left\|x^k - x(\lambda)\right\|^2\right] \leq \left(1 - \frac{\alpha\mu}{n}\right)^k \left\|x^0 - x(\lambda)\right\|^2 + \frac{2n\alpha\sigma^2}{\mu},$$
*where $\mathcal{L} := \frac{1}{n} \max\left\{\frac{L}{1-p}, \frac{\lambda}{p}\right\}$ and $\sigma^2 := \frac{1}{n^2} \sum_{i=1}^{n} \left(\frac{1}{1-p}\|\nabla f_i(x_i(\lambda))\|^2 + \frac{\lambda^2}{p}\|x_i(\lambda) - \overline{x}(\lambda)\|^2\right)$.*

Let us find the parameters $p$, $\alpha$ which lead to the fastest rate, to push the error within $\left(\mathcal{O}(\varepsilon) + \frac{2n\alpha\sigma^2}{\mu}\right)$-neighborhood of the optimum[5], i.e., to achieve
$$\mathbb{E}\left[\left\|x^k - x(\lambda)\right\|^2\right] \leq \varepsilon \left\|x^0 - x(\lambda)\right\|^2 + \frac{2n\alpha\sigma^2}{\mu}. \tag{10}$$

**Corollary 4.3** *The value $p^\star = \frac{\lambda}{L+\lambda}$ minimizes both the number of iterations and the expected number of communications for achieving (10). In particular, the optimal number of iterations is $2\frac{L+\lambda}{\mu} \log \frac{1}{\varepsilon}$, and the optimal expected number of communications is $\frac{2\lambda}{\lambda+L} \frac{L}{\mu} \log \frac{1}{\varepsilon}$.*

If we choose $p = p^\star$, then $\frac{\alpha\lambda}{np} = \frac{1}{2}$, and the aggregation rule (9) in Algorithm 1 becomes
$$x_i^{k+1} = \frac{1}{2}\left(x_i^k + \bar{x}^k\right) \tag{11}$$

---

[5]In Sec. 5 we propose a variance reduced algorithm which removes the $\left(\frac{2n\alpha\sigma^2}{\mu}\right)$-neighborhood from Thm. 4.2. In that setting, our goal will be to achieve $\mathbb{E}\left[\left\|x^k - x(\lambda)\right\|^2\right] \leq \varepsilon \left\|x^0 - x(\lambda)\right\|^2$.

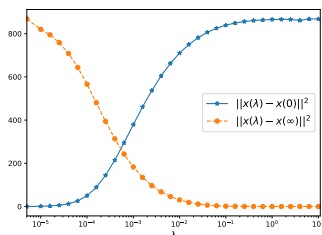
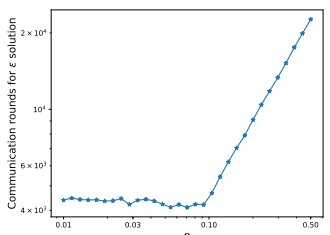

Figure 1: Distance of solution $x(\lambda)$ of (2) to pure local solution $x(0)$ and global solution $x(\infty)$ as a function of $\lambda$. Logistic regression on a1a dataset. See Appendix for the setup.

Figure 2: Communication rounds to get $\frac{F(x^k) - F(x^*)}{F(x^0) - F(x^*)} \leq 10^{-5}$ as a function of $p$ with $p^* \approx 0.09$ (for L2SGD+). Logistic regression on a1a dataset with $\lambda = 0.1$.

while the local GD step (8) becomes $x_i^{k+1} = x_i^k - \frac{1}{2L}\nabla f_i(x_i^k)$. Notice that while our method does not support full averaging as that is too unstable, (11) suggests that one should take a large step *towards* averaging. As $\lambda$ get smaller, the solution to the optimization problem (2) will increasingly favour pure local models, i.e., $x_i(\lambda) \to x_i(0) := \arg\min f_i$ for all $i$ as $\lambda \to 0$. Pure local models can be computed without any communication whatsoever and Cor. 4.3 confirms this intuition: the optimal number of communication round decreases to zero as $\lambda \to 0$. On the other hand, as $\lambda \to \infty$, the optimal number of communication rounds converges to $2\frac{L}{\mu}\log\frac{1}{\varepsilon}$, which recovers the performance of GD for finding the globally optimal model (see Fig. 1).

*In summary, we recover the communication efficiency of GD for finding the globally optimal model as $\lambda \to \infty$ (ignoring the $(\frac{2n\alpha\sigma^2}{\mu})$-neighborhood). However, for other values of $\lambda$, the communication complexity of L2GD is better and decreases to 0 as $\lambda \to 0$. Hence, our communication complexity result interpolates between the communication complexity of GD for finding the global model and the zero communication complexity for finding the pure local models.*

## 5 LOOPLESS LOCAL SGD WITH VARIANCE REDUCTION

As we have seen in Sec. 4.2, L2GD is a specific instance of SGD, thus only converges linearly to a neighborhood of the optimum. In this section, we resolve the mentioned issue by incorporating control variates to the stochastic gradient (Johnson & Zhang, 2013b; Defazio et al., 2014). We go further: we assume that each local objective has a finite-sum structure and propose an algorithm, L2SGD+, which takes *local stochastic* gradient steps, while maintaining (global) linear convergence rate. As a consequence, L2SGD+ is the first local SGD with linear convergence.[6] For convenience, we present variance reduced local GD (i.e., no local subsampling) in the Appendix.

**Assumption 5.1** *Assume that $f_i$ has a finite-sum structure: $f_i(x_i) = \frac{1}{m}\sum_{j=1}^m f'_{i,j}(x_i)$. Let $f'_{i,j}$ be convex, $L'$-smooth while $f_i$ is $\mu$-strongly convex (for each $1 \leq j \leq m, 1 \leq i \leq n$).*

### 5.1 CONVERGENCE THEORY

We are now ready to present a convergence rate of L2SGD+ (the algorithm, along with the efficient implementation is presented in Appendix C.4).

**Theorem 5.1** *Let Assumption 5.1 hold and choose $\alpha = n\min\left\{\frac{(1-p)}{4L'+\mu m}, \frac{p}{4\lambda+\mu}\right\}$. Then the iteration complexity of Algorithm 3 is $\max\left\{\frac{4L'+\mu m}{(1-p)\mu}, \frac{4\lambda+\mu}{p\mu}\right\}\log\frac{1}{\varepsilon}$.*

Next, we find the value of $p$ that yields both the best iteration and communication complexity.

---

[6]We are aware that a linearly converging local SGD (with $\lambda = \infty$) can be obtained as a particular instance of the decoupling method (Mishchenko & Richtárik, 2019). Other variance reduced local SGD algorithms (Liang et al., 2019; Karimireddy et al., 2019; Wu et al., 2019) do not achieve linear convergence.

**Corollary 5.2** *Both communication and iteration complexity of L2SGD+ are minimized for $p = \frac{4\lambda+\mu}{4\lambda+4L'+(m+1)\mu}$. The resulting iteration complexity is $\left(4\frac{\lambda}{\mu} + 4\frac{L'}{\mu} + m + 1\right)\log\frac{1}{\varepsilon}$, while the communication complexity is $\frac{4\lambda+\mu}{4L'+4\lambda+(m+1)\mu}\left(4\frac{L'}{\mu} + m\right)\log\frac{1}{\varepsilon}$.*

Note that with $\lambda \to \infty$, the communication complexity of L2SGD+ tends to $\left(4\frac{L'}{\mu} + m\right)\log\frac{1}{\varepsilon}$, which is communication complexity of minibatch SAGA to find the globally optimal model (Hanzely & Richtárik, 2019). On the other hand, in the pure local setting ($\lambda = 0$), the communication complexity becomes $\log\frac{1}{\epsilon}$ – this is because the Lyapunov function involves a term that measures the distance of local models, which requires communication to be estimated.

**Remark 5.3** *L2SGD+ is the simplest local SGD method with variance reduction. In the Appendix, we present L2SGD++ which allows for 1) an arbitrary number of data points per client and arbitrary local subsampling, 2) partial participation of clients, and 3) local SVRG-like updates of control variates (thus better memory). Lastly, L2SGD++ exploits the complex smoothness structure of the local objectives, resulting in tighter rates.*

## 6 EXPERIMENTS

In this section, we numerically verify the theoretical claims from this paper. We only present a single experiment here, all remaining ones along with the missing details about the setup are in the Appendix. In particular, the Appendix includes two more experiments. The first one studies how $p$ (communication) influences the convergence of L2SGD+. The second experiment aims to examine the effect of parameter $\lambda$ on the convergence rate of L2SGD+.

We consider logistic regression problem with LibSVM data (Chang & Lin, 2011). The data were normalized so that $f'_{i,j}$ is 1-smooth for each $j$, while the local objectives are $10^{-4}$-strongly convex. In order to cover a range of possible scenarios, we have chosen a different number of clients for each dataset (see the Appendix). Lastly, the stepsize was always chosen according to Thm. 5.1.

We compare three different methods: L2SGD+, L2GD with local subsampling (L2SGD in the Appendix), and L2GD with local subsampling and control variates constructed for $\psi$ only (L2SGD2 in the Appendix; similar to (Liang et al., 2019)). We expect L2SGD+ to converge to the global optimum linearly, while both L2SGD and L2SGD2 to converge to certain neighborhood. Each method is applied to two objectives constructed by a different split of the data among the devices. For the homogeneous split, we randomly reshuffle the data. For heterogeneous split, we first sort the data based on the labels and then construct the local objectives according to the current order.

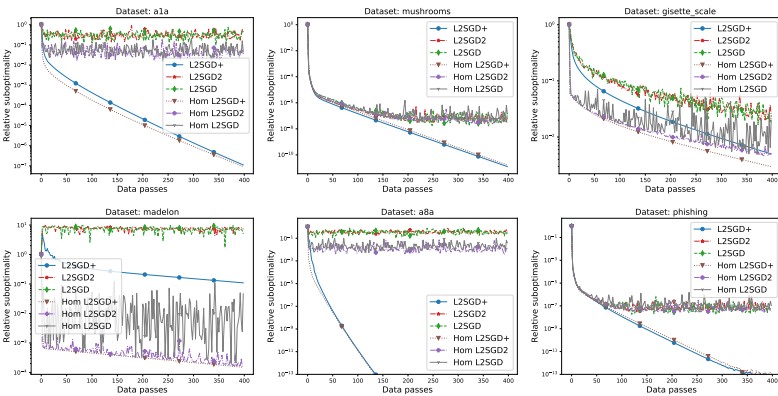

Figure 3: L2SGD+, vs L2SGD vs L2SGD2 with identical stepsize (details in the Appendix).

Fig. 3 demonstrates the importance of variance reduction – it ensures a fast global convergence of L2SGD+, while the neighborhood is slightly smaller for L2SGD2 compared to L2SGD. As predicted, data heterogeneity does not affect the convergence speed of the proposed methods.

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

# Appendix
# Federated Learning of a Mixture of Global and Local Models

CONTENTS

## A    POSSIBLE EXTENSIONS

Our analysis of L2GD can be extended to cover smooth convex and non-convex loss functions $f_i$ (we do not explore these directions). Further, our methods can be extended to a decentralized regime where the devices correspond to devices of a connected network, and communication is allowed along the edges of the graph only. This can be achieved by introducing an additional randomization over the penalty $\psi$. Further, our approach can be accelerated in the sense of Nesterov (Nesterov, 2004) by adapting the a variant of Katyusha (Allen-Zhu, 2017; Qian et al., 2019a) to our setting, thus further reducing the number of communication rounds.

## B    EXPERIMENTAL SETUP AND FURTHER EXPERIMENTS

In all experiments in this paper, we consider a simple binary classification model – logistic regression. In particular, suppose that device $i$ owns data matrix $\mathbf{A}_i \in \mathbb{R}^{m \times d}$ along with corresponding labels $b_i \in \{-1, 1\}^m$. The local objective for client $i$ is then given as follows

$$f_i(x) := \frac{1}{m} \sum_{j=1}^{m} f'_{i,j}(x) + \frac{\mu}{2}\|x\|^2, \quad \text{where} \quad f'_{im+j}(x) = \log\left(1 + \exp\left((\mathbf{A}_i)_{j,:}x \cdot b_i\right)\right).$$

The rows of data matrix $\mathbf{A}$ were normalized to have length 4 so that each $f'_{i,j}$ is 1-smooth for each $j$. At the same time, the local objective on each device is $10^{-4}$ strongly convex. Next, datasets are from LibSVM (Chang & Lin, 2011).

In each case, we consider the simplest locally stochastic algorithm. In particular, each dataset is evenly split among the clients, while the local stochastic method samples a single data point each iteration.

We have chosen a different number of clients for each dataset – so that we cover different possible scenarios. See Table 1 for details (it also includes sizes of the datasets). Lastly, the stepsize was always chosen according to Thm. 5.1.

Table 1: Setup for the experiments.

| Dataset | $N$ $= nm$ | $d$ | $n$ | $m$ | $\mu$ | $L$ | $p$ (Sec. B.1) | $\lambda$ (Sec. B.2) | $p$ (Sec. B.3) |
|---|---|---|---|---|---|---|---|---|---|
| a1a | 1 605 | 123 | 5 | 321 | $10^{-4}$ | 1 | 0.1 | 0.1 | 0.1 |
| mushrooms | 8 124 | 112 | 12 | 677 | $10^{-4}$ | 1 | 0.1 | 0.05 | 0.3 |
| phishing | 11 055 | 68 | 11 | 1 005 | $10^{-4}$ | 1 | 0.1 | 0.1 | 0.001 |
| madelon | 2 000 | 500 | 50 | 40 | $10^{-4}$ | 1 | 0.1 | 0.02 | 0.05 |
| duke | 44 | 7 129 | 4 | 11 | $10^{-4}$ | 1 | 0.1 | 0.4 | 0.1 |
| gisette_scale | 6 000 | 5 000 | 100 | 60 | $10^{-4}$ | 1 | 0.1 | 0.2 | 0.003 |
| a8a | 22 696 | 123 | 8 | 109 | $10^{-4}$ | 1 | 0.1 | 0.1 | 0.1 |

### B.1    COMPARISON OF THE METHODS

In our first experiment, we verify two phenomena:

- The effect of variance reduction on the convergence speed of local methods. We compare 3 different methods: local SGD with full variance reduction (Algorithm 3), shifted local SGD (Algorithm 7) and local SGD (Algorithm 6). Our theory predicts that a fully variance reduced algorithm converges to the global optimum linearly, while both shifted local SGD and local SGD converge to a neighborhood of the optimum. At the same time, the neighborhood should be smaller for shifted local SGD.

- The claim that heterogeneity of the data does not influence the convergence rate. We consider two splits of the data heterogeneous and homogeneous. For the homogeneous split, we first randomly reshuffle the data and then construct the local objectives according to the current order (i.e., the first client owns the first $m$ indices, etc.). For heterogeneous split, we first sort the data based on the labels and then construct the local objectives accordingly (thus achieving the worst-case heterogeneity). Note that the overall objective to solve is

different in homogeneous and heterogeneous case – we thus plot relative suboptimality of the objective (i.e., $\frac{F(x^k)-F(x^\star)}{F(x^0)-F(x^\star)}$) to directly compare the convergence speed.

In each experiment, we choose $p = 0.1$ and $\lambda = \frac{1}{9}$ – such choice mean that $p$ is very close to optimal. The other parameters (i.e., number of clients) are provided in Table 1. Fig. 4 presents the result.

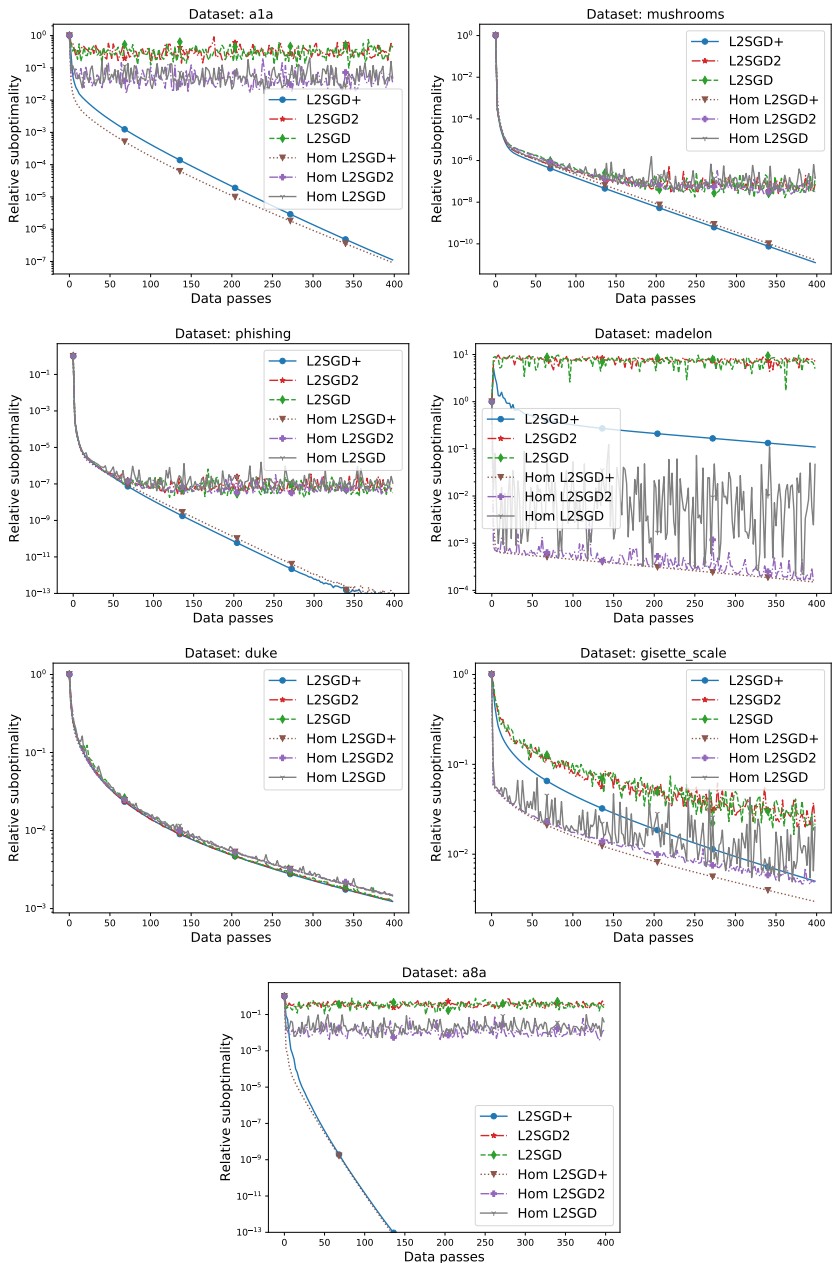

Figure 4: Variance reduced local SGD (Algorithm 3), shifted local SGD (Algorithm 7) and local SGD (Algorithm 6) applied on LibSVM problems for both homogeneous split of data and Heterogeneous split of the data. Stepsize for non-variance reduced method was chosen the same as for the analogous variance reduced method.

As expected, Figure 4 clearly demonstrates the following:

- Full variance reduction always converges to the global optima, methods with partial variance reduction only converge to a neighborhood of the optimum.

- Partial variance reduction (i.e., shifting the local SGD) is better than not using control variates at all. Although the improvement in the performance is rather negligible.

- Data heterogeneity does not affect the convergence speed of the proposed methods. Therefore, unlike standard local SGD, mixing the local and global models does not suffer the problems with heterogeneity.

### B.2 EFFECT OF $p$

In the second experiment, we study the effect of $p$ on the convergence rate of variance reduced local SGD. Note that $p$ immediately influences the number of communication rounds – on average, the clients take $(p^{-1} - 1)$ local steps in between two consecutive rounds of communication (aggregation).

In Section 5, we argue that, it is optimal (in terms of the convergence rate) to choose $p$ of order $p^\star := \frac{\lambda}{L' + \lambda}$. Figure 5 compares $p = p^\star$ against other values of $p$ and confirms its optimality (in terms of optimizing the convergence rate).

While the slower convergence of Algorithm 3 with $p < p^\star$ is expected (i.e., communicating more frequently yields a faster convergence), slower convergence for $p > p^\star$ is rather surprising; in fact, it means that communicating less frequently yields faster convergence. This effect takes place due to the specific structure of problem (12); it would be lost when enforcing $x_1 = \cdots = x_n$ (corresponding to $\lambda = \infty$).

### B.3 EFFECT OF $\lambda$

In this experiment we study how different values of $\lambda$ influence the convergence rate of Algorithm 3, given that everything else (i.e., $p$) is fixed. Note that for each value of $\lambda$ we get a different instance of problem (12); thus the optimal solution is different as well. Therefore, in order to make a fair comparison between convergence speeds, we plot the relative suboptimality (i.e., $\frac{F(x^k) - F(x^\star)}{F(x^0) - F(x^\star)}$) against the data passes. Figure 6 presents the results.

The complexity of Algorithm 3 is[7] $\mathcal{O}\left(\frac{L'}{(1-p)\mu}\right) \log \frac{1}{\varepsilon}$ as soon as $\lambda < \lambda^\star := \frac{Lp}{(1-p)}$; otherwise the complexity is $\mathcal{O}\left(\frac{\lambda}{p\mu}\right) \log \frac{1}{\varepsilon}$. This perfectly consistent with what Figure 6 shows – the choice $\lambda < \lambda^\star$ resulted in comparable convergence speed than $\lambda = \lambda^\star$; while the choice $\lambda > \lambda^\star$ yields noticeably worse rate than $\lambda = \lambda^\star$.

---

[7] Given that $\mu$ is small.

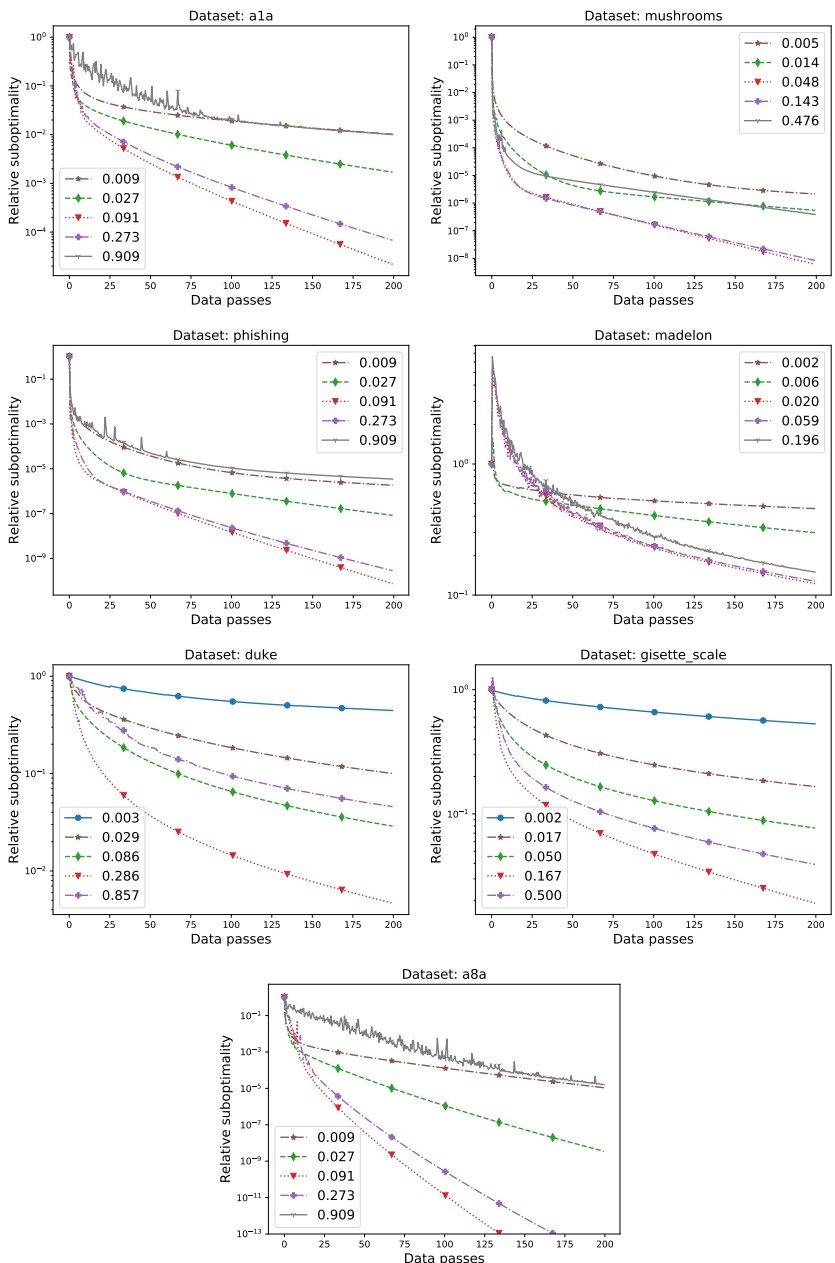

Figure 5: Effect of the aggregation probability $p$ (legend of the plots) on the convergence rate of Algorithm 3. Choice $p = p^\star$ corresponds to red dotted line with triangle marker. Parameter $\lambda$ was chosen in each case as Table 1 indicates.

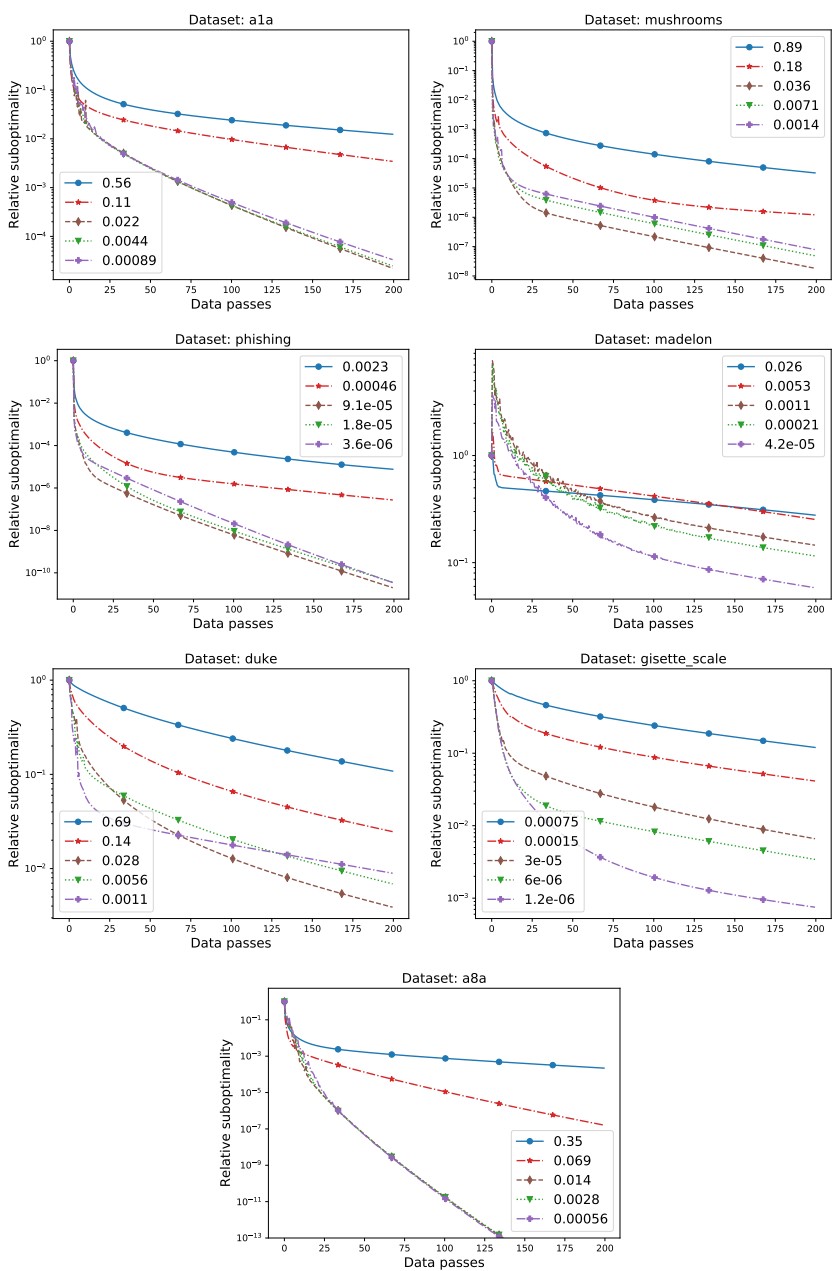

Figure 6: Effect of parameter $\lambda$ (legend of the plot) on the convergence rate of Algorithm 3. The choice $\lambda = \lambda^\star$ corresponds to borwn dash-dotted line with diamond marker (the third one from the legend). Aggregation probability $p$ was chosen in each case as Table 1 indicates.

# C    REMAINING ALGORITHMS

## C.1    UNDERSTANDING COMMUNICATION OF L2GD

**Example C.1** *In order to better understand when communication takes place in Algorithm 1, consider the following possible sequence of coin tosses: $0, 0, 1, 0, 1, 1, 1, 0$. The first two coin tosses lead to two local GD steps* (8) *on all devices. The third coin toss lands $1$, at which point all local models $x_i^k$ are communicated to the master, averaged to form $\bar{x}^k$, and the step* (9) *towards averaging is taken. The fourth coin toss is $0$, and at this point, the master communicates the updated local models back to the devices, which subsequently perform a single local GD step* (8)*. Then come three consecutive coin tosses landing $1$, which means that the local models are again communicated to the master, which performs three averaging steps* (9)*. Finally, the eighth coin toss lands $0$, which makes the master send the updated local models back to the devices, which subsequently perform a single local GD step.*

This example illustrates that communication needs to take place whenever two consecutive coin tosses land a different value. If $0$ is followed by a $1$, all devices communicate to the master, and if $1$ is followed by a $0$, the master communicates back to the devices. It is standard to count each pair of communications, Device→Master and the subsequent Master→Device, as a single communication round.

**Lemma C.2** *The expected number of communication rounds in $k$ iterations of L2GD is $p(1 - p)k$.*

## C.2    L2GD AND FULL AVERAGING

Is a setup such that conditions of Thm. 4.2 are satisfied and the aggregation update (9) is identical to full averaging? This is equivalent requiring $0 < p < 1$ such that $\alpha\lambda = np$. However, we have $\alpha\lambda \leq \frac{\lambda}{2\mathcal{L}} \leq np$, which means that full averaging is not supported by our theory.

## C.3    LOCAL GD WITH VARIANCE REDUCTION

In this section, we present variance reduced local gradient descent with partial aggregation. In particular, the proposed algorithm (Algorithm 2) incorporates control variates to Algorithm 1. Therefore, the proposed method can be seen as a special case of Algorithm 3 with $m = 1$. We thus only present it for pedagogical purposes, as it might shed additional insights into our approach.

In particular, the update rule of proposed method will be $x^{k+1} = x^k - \alpha g^k$ where

$$g^k = \begin{cases} p^{-1}(\lambda\nabla\psi(x^k) - n^{-1}\boldsymbol{\Psi}^k) + n^{-1}\mathbf{J}^k + n^{-1}\boldsymbol{\Psi}^k & \text{with probability} \quad p \\ (1-p)^{-1}(\nabla f(x^k) - n^{-1}\mathbf{J}^k) + n^{-1}\mathbf{J}^k + n^{-1}\boldsymbol{\Psi}^k & \text{with probability} \quad 1 - p \end{cases}.$$

for some control variates vectors $\mathbf{J}^k, \boldsymbol{\Psi}^k \in \mathbb{R}^{nd}$. A quick check gives

$$\mathbb{E}\left[g^k \,|\, x^k\right] = \nabla f(x^k) + \lambda\nabla\psi(x^k) = \nabla F(x^k),$$

thus the direction we are taking is unbiased regardless of the value of control variates $\mathbf{J}^k, \boldsymbol{\Psi}^k$. The goal is to make control variates $\mathbf{J}^k, \boldsymbol{\Psi}^k$ correlated[8] with $n\nabla f(x^k)$ and $n\lambda\nabla\psi(x^k)$. One possible solution to the problem is for $\mathbf{J}^k, \boldsymbol{\Psi}^k$ to track most recently observed values of $n\nabla f(\cdot)$ and $n\lambda\nabla\psi(\cdot)$, which corresponds to the following update rule

$$\left(\boldsymbol{\Psi}^{k+1}, \mathbf{J}^{k+1}\right) = \begin{cases} \left(n\lambda\nabla\psi(x^k), \mathbf{J}^k\right) & \text{with probability} \quad p \\ \left(\boldsymbol{\Psi}^k, n\nabla f(x^k)\right) & \text{with probability} \quad 1 - p \end{cases}.$$

A specific, distributed implementation of the described method is presented as Algorithm 2. The only communication between the devices takes place when the average model $\bar{x}^k$ is being computed (with probability $p$), which is analogous to standard local SGD. Therefore we aim to set $p$ rather small.

Note that Algorithm 2 is a particular special case of SAGA with importance sampling (Qian et al., 2019b); thus, we obtain convergence rate of the method for free. We state it as Thm. C.3.

---

[8]Specifically we aim to have Corr $\left[\mathbf{J}^k, n\nabla f(x^k)\right] \to 1$ and Corr $\left[n^{-1}\boldsymbol{\Psi}^k, \lambda\nabla\psi(x^k)\right] \to 1$ as $x^k \to x^\star$.

---

**Algorithm 2** Variance reduced local gradient descent

---

**Input:** $x_1^0 = \cdots = x_n^0 \in \mathbb{R}^d$, stepsize $\alpha$, probability $p$
$\mathbf{J}_1^0 = \cdots = \mathbf{J}_n^0 = \mathbf{\Psi}_1^0 = \cdots = \mathbf{\Psi}_n^0 = 0 \in \mathbb{R}^d$
**for** $k = 0, 1, \ldots$ **do**
$\quad \xi = 1$ with probability $p$ and $0$ with probability $1 - p$
$\quad$**if** $\xi$ **then**
$\quad\quad$All Devices $i = 1, \ldots, n$:
$\quad\quad\quad$Compute $\nabla f_i(x_i^k)$
$\quad\quad\quad x_i^{k+1} = x_i^k - \alpha \left( n^{-1}(1-p)^{-1} \nabla f_i(x_i^k) - n^{-1} \frac{p}{1-p} \mathbf{J}_i^k + n^{-1} \mathbf{\Psi}_i^k \right)$
$\quad\quad\quad$Set $\mathbf{J}_i^{k+1} = \nabla f_i(x_i^k)$, $\mathbf{\Psi}_i^{k+1} = \mathbf{\Psi}_i^k$
$\quad$**else**
$\quad\quad$Master computes the average $\bar{x}^k = \frac{1}{n} \sum_{i=1}^n x_i^k$
$\quad\quad$Master does for all $i = 1, \ldots, n$:
$\quad\quad\quad$Set $x_i^{k+1} = x_i^k - \alpha \left( \frac{\lambda}{np}(x_i^k - \bar{x}^k) - (p^{-1} - 1) n^{-1} \mathbf{\Psi}_i^k + n^{-1} \mathbf{J}_i^k \right)$
$\quad\quad\quad$Set $\mathbf{\Psi}_i^{k+1} = \lambda(x_i^k - \bar{x}^k)$, $\mathbf{J}_i^{k+1} = \mathbf{J}_i^k$
$\quad$**end if**
**end for**

---

**Theorem C.3** *Let Assumption 3.1 hold. Set $\alpha = n \min \left( \frac{(1-p)}{4L+\mu}, \frac{p}{4\lambda+\mu} \right)$. Then, iteration complexity of Algorithm 2 is*

$$\max \left( \frac{4L+\mu}{\mu(1-p)}, \frac{4\lambda+\mu}{\mu p} \right) \log \frac{1}{\varepsilon}.$$

**Proof:** Clearly,

$$F(x) = f(x) + \lambda\psi(x) = \frac{1}{2} \left( \underbrace{2f(x)}_{:= \mathtt{f}(x)} + \underbrace{2\lambda\psi(x)}_{:= \psi(x)} \right).$$

Note that $\psi$ is $\frac{2\lambda}{n}$ smooth and $\mathtt{f}$ is $\frac{2L}{n}$ smooth. At the same time, $F$ is $\frac{\mu}{n}$ strongly convex. Using convergence theorem of SAGA with importance sampling from (Qian et al., 2019b; Gazagnadou et al., 2019), we get

$$\mathbb{E}\left[ F(x^k) + \frac{\alpha}{2} \Upsilon(\mathbf{J}^k, \mathbf{\Psi}^k) \right] \leq \left( 1 - \alpha\frac{\mu}{n} \right)^k \left( F(x^0) + \frac{\alpha}{2} \Upsilon(\mathbf{J}^0, \mathbf{\Psi}^0) \right),$$

where

$$\Upsilon(\mathbf{J}^k, \mathbf{\Psi}^k) := \frac{4}{n^2} \sum_{i=1}^n \left( \left\| \mathbf{\Psi}_i^k - \lambda(x_i(\lambda) - \bar{x}(\lambda)) \right\|^2 + \left\| \mathbf{J}_i^k - \nabla f_i(x_i(\lambda)) \right\|^2 \right)$$

and $\alpha = n \min \left( \frac{(1-p)}{4L+\mu}, \frac{p}{4\lambda+\mu} \right)$, as desired.

**Corollary C.4** *Iteration complexity of Algorithm 2 is minimized for $p = \frac{4\lambda+\mu}{4\lambda+4L+2\mu}$, which yields complexity $4 \left( \frac{\lambda}{\mu} + \frac{L}{\mu} + \frac{1}{2} \right) \log \frac{1}{\varepsilon}$. The communication complexity is minimized for any $p \leq \frac{4\lambda+\mu}{4\lambda+4L+2\mu}$, in which case the total number of communication rounds to reach $\varepsilon$-solution is $\left( \frac{4\lambda}{\mu} + 1 \right) \log \frac{1}{\varepsilon}$.*

As a direct consequence of Corollary C.4 we see that the optimal choice of $p$ that minimizes both communication and number of iterations to reach $\varepsilon$ solution of problem (17) is $p = \frac{4\lambda+\mu}{4\lambda+4L+2\mu}$.

**Remark C.5** *While both Algorithm 2 and Algorithm 3 are a special case of SAGA, the practical version of variance reduced local SGD (presented in Section C.5) is not. In particular, we wish to run the SVRG-like method locally in order to avoid storing the full gradient table.[9] Therefore,*

---

[9] SAGA does not require storing a full gradient table for problems with linear models by memorizing the residuals. However, in full generality, SVRG-like methods are preferable.

*variance reduced local SGD that will be proposed in Section C.5 is neither a special case of SAGA nor a special case of SVRG (or a variant of SVRG). However, it is still a special case of a more general algorithm from (Hanzely & Richtárik, 2019).*

As mentioned, Algorithm 3 is a generalization of Algorithm 2 when the local subproblem is a finite sum. Note that Algorithm 2 constructs a control variates for both local subproblem and aggregation function $\psi$ and constructs corresponding unbiased gradient estimator. In contrast, Algorithm 3 constructs extra control variates within the local subproblem in order to reduce the variance of gradient estimator coming from the local subsampling.

## C.4 L2SGD+: ALGORITHM AND THE EFFICIENT IMPLEMENTATION

Denote $\mathbf{1} \in \mathbb{R}^m$ to be vector of ones. We are now ready to state L2SGD+ as Algorithm 3.

---

**Algorithm 3** L2SGD+: Loopless Local SGD with Variance Reduction

---

**Input:** $x_1^0 = \cdots = x_n^0 \in \mathbb{R}^d$, stepsize $\alpha$, probability $p$
$\mathbf{J}_i^0 = 0 \in \mathbb{R}^{d \times m}, \mathbf{\Psi}_i^0 = 0 \in \mathbb{R}^d$ (for $i = 1, \ldots, n$)
**for** $k = 0, 1, \ldots$ **do**
 $\xi = 1$ with probability $p$ and 0 with probability $1 - p$
 **if** $\xi = 0$ **then**
  All Devices $i = 1, \ldots, n$:
   Sample $j \in \{1, \ldots, m\}$ (uniformly at random)
   $g_i^k = \frac{1}{n(1-p)} \left( \nabla f'_{i,j}(x_i^k) - \left( \mathbf{J}_i^k \right)_{:,j} \right) + \frac{\mathbf{J}_i^k \mathbf{1}}{nm} + \frac{\mathbf{\Psi}_i^k}{n}$
   $x_i^{k+1} = x_i^k - \alpha g_i^k$
   Set $(\mathbf{J}_i^{k+1})_{:,j} = \nabla f'_{i,j}(x_i^k), \mathbf{\Psi}_i^{k+1} = \mathbf{\Psi}_i^k,$
    $(\mathbf{J}_i^{k+1})_{:,l} = (\mathbf{J}_i^{k+1})_{:,l}$ for all $l \neq j$
 **else**
  Master computes the average $\bar{x}^k = \frac{1}{n} \sum_{i=1}^n x_i^k$
  Master does for all $i = 1, \ldots, n$:
   $g_i^k = \frac{\lambda}{np}(x_i^k - \bar{x}^k) - \frac{p^{-1}-1}{n}\mathbf{\Psi}_i^k + \frac{1}{nm}\mathbf{J}_i^k \mathbf{1}$
   Set $x_i^{k+1} = x_i^k - \alpha g_i^k$
   Set $\mathbf{\Psi}_i^{k+1} = \lambda(x_i^k - \bar{x}^k), \mathbf{J}_i^{k+1} = \mathbf{J}_i^k$
 **end if**
**end for**

---

L2SGD+ only communicates when a two consecutive coin tosses land a different value, thus, on average $p(1 - p)k$ times per $k$ iterations. However, L2SGD+ requires communication of control variates $\mathbf{J}_i \mathbf{1}, \mathbf{\Psi}_i$ as well – each communication round is thus three times more expensive. In the Appendix, we provide an implementation of L2SGD+ that does not require the communication of $\mathbf{J}_i \mathbf{1}, \mathbf{\Psi}_i$.

Here we present an efficient implementation of L2SGD+ as Algorithm 4 so that we do not have to communicate control variates. As a consequence, Algorithm 4 needs to communicate on average $p(1 - p)k$ times per $k$ iterations, while each communication consists of sending only local models to the master and back.

## C.5 LOCAL SGD WITH VARIANCE REDUCTION – GENERAL METHOD

In this section, we present a fully general variance reduced local SGD. We consider a more general instance of (2) where each local objective includes a possibly nonsmooth regularizer, which admits a cheap evaluation of proximal operator. In particular, the objective becomes

---

**Algorithm 4** L2SGD+: Loopless Local SGD with Variance Reduction (communication-efficient implementation)

---

**Input:** $x_1^0 = \cdots = x_n^0 = \tilde{x} \in \mathbb{R}^d$, stepsize $\alpha$, probability $p$
Initialize control variates $\mathbf{J}_i^0 = 0 \in \mathbb{R}^{d \times m}, \mathbf{\Psi}_i^0 = 0 \in \mathbb{R}^d$ (for $i = 1, \ldots, n$), initial coin toss $\xi^{-1} = 0$
**for** $k = 0, 1, \ldots$ **do**
  $\xi^k = 1$ with probability $p$ and $0$ with probability $1 - p$
  **if** $\xi^k = 0$ **then**
  All Devices $i = 1, \ldots, n$:
    **if** $\xi^{k-1} = 1$ **then**
      Receive $x_i^k, c$ from Master
      Reconstruct $\bar{x}^k = \bar{x}^{k-c}$ using $x_i^k, x_i^{k-c}, c$
      Set $x_i^k = x_i^k - c\alpha \frac{1}{nm} \mathbf{J}_i^k \mathbf{1}, \mathbf{J}_i^k = \mathbf{J}_i^{k-c}, \mathbf{\Psi}_i^k = \lambda(x_i^{k-c} - \bar{x}^k)$,
    **end if**
    Sample $j \in \{1, \ldots, m\}$ (uniformly at random)
    $g_i^k = \frac{1}{n(1-p)} \left( \nabla f'_{i,j}(x_i^k) - \left(\mathbf{J}_i^k\right)_{:,j} \right) + \frac{\mathbf{J}_i^k \mathbf{1}}{nm} + \frac{\mathbf{\Psi}_i^k}{n}$
    $x_i^{k+1} = x_i^k - \alpha g_i^k$
    Set $(\mathbf{J}_i^{k+1})_{:,j} = \nabla f'_{i,j}(x_i^k), \mathbf{\Psi}_i^{k+1} = \mathbf{\Psi}_i^k$,
      $(\mathbf{J}_i^{k+1})_{:,l} = (\mathbf{J}_i^{k+1})_{:,l}$ for all $l \neq j$
  **else**
  Master does for all $i = 1, \ldots, n$:
    **if** $\xi^{k-1} = 0$ **then**
      Set $c = 0$
      Receive $x_i^k$ from Device and set $\bar{x} = \frac{1}{n} \sum_{i=1}^n x_i^k, x_i^k = x_i^k$
    **end if**
    Set $x_i^{k+1} = x_i^k - \alpha \left( \frac{\lambda}{np}(x_i^k - \bar{x}) - \frac{p^{-1}-1}{n}\lambda(\tilde{x} - \bar{x}) \right)$
    Set $\tilde{x} = x_i^k$
    Set $c = c + 1$
  **end if**
**end for**

---

$$\min_{x \in \mathbb{R}^{dn}} \frac{1}{N} \sum_{i=1}^n \underbrace{\left( \underbrace{\sum_{j=1}^{m_i} f'_{i,j}(x_i)}_{=\frac{N}{n} f_i(x)} \right)}_{=f(x)} + \lambda \underbrace{\frac{1}{2n} \sum_{i=1}^n \|x_i - \bar{x}\|^2}_{=\psi(x)} + \underbrace{\sum_{i=1}^n R_i(x_i)}_{:=R(x)}, \qquad (12)$$

$$\underbrace{\phantom{\min_{x} \frac{1}{N} \sum_{i=1}^n \sum_{j=1}^{m_i} f'_{i,j}(x_i) + \lambda \frac{1}{2n} \sum_{i=1}^n}}_{=F(x)}$$

where $m_i$ is the number of data points owned by client $i$ and $N = \sum_{i=1}^n m_i$.

In order to squeeze a faster convergence rate from minibatch samplings, we will assume that $f'_{i,j}$ is smooth with respect to a matrix $\mathbf{M}_{i,j}$ (instead of scalar $L'_{i,j} = \lambda_{\max} \mathbf{M}_{i,j}$).

**Assumption C.1** *Suppose that $f'_{i,j}$ is $\mathbf{M}_{i,j}$ smooth ($\mathbf{M}_{i,j} \in \mathbb{R}^{d \times d}, \mathbf{M}_{i,j} \succ 0$) and $\mu$ strongly convex for $1 \leq j \leq m_i, 1 \leq i \leq n$, i.e.*

$$f'_{i,j}(y) + \langle \nabla f'_{i,j}(y), x - y \rangle \leq f'_{i,j}(x) \leq f'_{i,j}(y) + \langle \nabla f'_{i,j}(y), x - y \rangle + \frac{1}{2} \|y - x\|^2_{\mathbf{M}_{i,j}}, \quad \forall x, y \in \mathbb{R}^d. \tag{13}$$

*Furthermore, assume that $R_i$ is convex for $1 \leq i \leq n$.*

Our method (Algotihm 5) allows for arbitrary aggregation probability (same as Algorithms 2, 3), arbitrary sampling of clients (to model the inactive clients) and arbitrary structure/sampling of the local objectives (i.e., arbitrary size of local datasets, arbitrary smoothness structure of each local

objective and arbitrary subsampling strategy of each client). Moreover, it allows for the SVRG-like update rule of local control variates $\mathbf{J}^k$, which requires less storage given an efficient implementation.

To be specific, each device owns a distribution $\mathcal{D}_i$ over subsets of $m_i$. When the aggregation is not performed (with probability $1 - p$), a subset of active devices $S$ is selected ($S$ follows arbitrary fixed distribution $\mathcal{D}$). Each of the active clients ($i \in S$) samples a subset of local indices $S_i \sim \mathcal{D}_i$ and observe the corresponding part of local Jacobian $\mathbf{G}_i(x^k)_{(:,S_i)}$ (where $\mathbf{G}_i(x^k) := [\nabla f'_{i,1}(x^k), \nabla f'_{i,2}(x^k), \dots \nabla f'_{i,m_i}(x^k))$. When the aggregation is performed (with probability $p$) we evaluate $\bar{x}^k$ and distribute it to each device; using which each device computes a corresponding component of $\lambda \nabla \psi(x^k)$. Those are the key components in constructing the unbiased gradient estimator (without control variates).

It remains to construct control variates and unbiased gradient estimator. If the aggregation is done, we just simply replace the last column of the gradient table. If the aggregation is not done, we have two options – either keep replacing the columns of the Jacobian table (in such case, we obtain a particular case of SAGA (Defazio et al., 2014)) or do LSVRG-like replacement (Hofmann et al., 2015; Kovalev et al., 2020) (in such case, the algorithm is a particular case of GJS (Hanzely & Richtárik, 2019), but is not a special case of neither SAGA nor LSVRG. Note that LSVRG-like replacement is preferrable in practice due to a better memory efficiency (one does not need to store the whole gradient table) for the models other than linear.

In order to keep the gradient estimate unbiased, it will be convenient to define vector $p_i \in \mathbb{R}^{m_i}$ such that for each $j \in \{1, \dots, m_i\}$ we have $\mathbb{P}(j \in S_i) = p_{i,j}$.

Next, to give a tight rate for any given pair of smoothness structure and sampling strategy, we use a standard tool first proposed for the analysis of randomized coordinate descent methods (Richtárik & Takáč, 2016; Qu & Richtárik, 2016) called *Expected Separable Overapproximation (ESO)* assumption. ESO provides us with smoothness parameters of the objective which "account" for the given sampling strategy.

**Assumption C.2** *Suppose that there is $v_i \in \mathbb{R}^{m_i}$ such for each client we have:*

$$\mathbb{E}\left[\left\|\sum_{j \in S_i} \mathbf{M}_{i,j}^{\frac{1}{2}} h_{i,j}\right\|^2\right] \leq \sum_{j=1}^{m_i} p_{i,j} v_{i,j} \|h_{i,j}\|^2, \qquad \forall\, 1 \leq i \leq n, \forall h_{i,j} \in \mathbb{R}^{m_i}, j \in \{1, \dots, m_i\}.$$

(14)

Lastly, denote $p_i$ to be the probability that worker $i$ is active and $\mathbf{1}^{(m_i)} \in \mathbb{R}^{m_i}$ to be the vector of ones. The resulting algorithm is stated as Algorithm 5.

Next, Theorems C.6 and C.7 present convergence rate of Algorithm 5 (SAGA and SVRG variant, respectively).

**Theorem C.6** *Suppose that Assumptions C.1 and C.2 hold. Let*

$$\alpha = \min\left\{\min_{j \in \{1, \dots, m_i\}, 1 \leq i \leq n} \frac{N(1-p)p_{i,j}p_i}{4v_j + N\frac{\mu}{n}}, \frac{np}{4\lambda + \mu}\right\}.$$

*Then the iteration complexity of Algorithm 5 (SAGA option) is*

$$\max\left\{\max_{j \in \{1, \dots, m_i\}, 1 \leq i \leq n}\left(\frac{4v_j\frac{n}{N} + \mu}{\mu(1-p)p_{i,j}p_i}\right), \frac{4\lambda + \mu}{p\mu}\right\}\log\frac{1}{\varepsilon}.$$

**Theorem C.7** *Suppose that Assumptions C.1 and C.2 hold. Let*

$$\alpha = \min\left\{\min_{j \in \{1, \dots, m_i\}, 1 \leq i \leq n} \frac{N(1-p)p_i}{4\frac{v_j}{p_{i,j}} + N\frac{\mu}{n}p_i^{-1}}, \frac{pn}{4\lambda + \mu}\right\}.$$

*Then the iteration complexity of Algorithm 5 (LSVRG option) is*

$$\max\left\{\max_{j \in \{1, \dots, m_i\}, 1 \leq i \leq n}\left(\frac{4v_j\frac{n}{Np_{i,j}} + \mu p_i^{-1}}{p_i\mu(1-p)}\right), \frac{4\lambda + \mu}{p\mu}\right\}\log\frac{1}{\varepsilon}.$$

---

**Algorithm 5** L2SGD++: Loopless Local SGD with Variance Reduction and Partial Participation

---

**Input:** $x_1^0, \ldots x_n^0 \in \mathbb{R}^d$, # parallel units $n$, each of them owns $m_i$ data points (for $1 \leq i \leq n$), distributions $\mathcal{D}_t$ over subsets of $\{1, \ldots, m_i\}$, distribution $\mathcal{D}$ over subsets of $\{1, 2, \ldots n\}$, aggregation probability $p$, stepsize $\alpha$
$\mathbf{J}_i^0 = 0 \in \mathbb{R}^{d \times m_i}, \mathbf{\Psi}_i^0 = 0 \in \mathbb{R}^d$ (for $i = 1, \ldots, n$)
**for** $k = 0, 1, \ldots$ **do**
  $\xi = 1$ with probability $p$ and $0$ with probability $1 - p$
  **if** $\xi = 0$ **then**
    Sample $S \sim \mathcal{D}$
    All Devices $i \in S$:
      Sample $S_i \sim \mathcal{D}_i$; $S_i \subseteq \{1, \ldots, m_i\}$ (independently on each machine)
      Observe $\nabla f'_{i,j}(x_i^k)$ for all $j \in S_i$
      $g_i^k = \frac{1}{N(1-p)p_i} \left( \sum_{j \in S_i} p_{i,j}^{-1} \left( \nabla f'_{i,j}(x_i^k) - \left( \mathbf{J}_i^k \right)_{:,j} \right) \right) + \frac{1}{N} \mathbf{J}_i^k \mathbf{1}^{(m_i)} + n^{-1} \mathbf{\Psi}_i^k$
      $x_i^{k+1} = \text{prox}_{\alpha R_i}(x_i^k - \alpha g_i^k)$

      For all $j \in \{1, \ldots, m_i\}$ set $\mathbf{J}_{:,j}^{k+1} = \begin{cases} \begin{cases} \nabla f'_{i,j}(x_i^k) & \text{if} \quad j \in S_i \\ \mathbf{J}_{:,j}^k & \text{otherwise} \end{cases} & \text{if} \quad \text{SAGA} \\ \begin{cases} \nabla f'_{i,j}(x_i^k); & \text{w. p.} \quad p_i \\ \mathbf{J}_{:,j}^k & \text{otherwise} \end{cases} & \text{if} \quad \text{L} - \text{SVRG} \end{cases}$

      Set $\mathbf{\Psi}_i^{k+1} = \mathbf{\Psi}_i^k$
    All Devices $i \notin S$:
      $g_i^k = \frac{1}{N} \mathbf{J}_i^k \mathbf{1}^{(m_i)} + n^{-1} \mathbf{\Psi}_i^k$
      $x_i^{k+1} = \text{prox}_{\alpha R_i}(x_i^k - \alpha g_i^k)$
      Set $\mathbf{J}_i^{k+1} = \mathbf{J}_i^k, \mathbf{\Psi}_i^{k+1} = \mathbf{\Psi}_i^k$
  **else**
    Master computes the average $\bar{x}^k = \frac{1}{n} \sum_{i=1}^n x_i^k$
    Master does for all $i = 1, \ldots, n$:
      $g_i^k = p^{-1}\lambda(x_i^k - \bar{x}^k) - (p^{-1} - 1)n^{-1}\mathbf{\Psi}_i^k + \frac{1}{N} \mathbf{J}_i^k \mathbf{1}^{(m_i)}$
      Set $x_i^{k+1} = \text{prox}_{\alpha R_i} \left( x_i^k - \alpha g_i^k \right)$
      Set $\mathbf{\Psi}_i^{k+1} = \lambda(x_i^k - \bar{x}^k), \mathbf{J}_i^{k+1} = \mathbf{J}_i^k$
  **end if**
**end for**

---

**Remark C.8** *Algotihm 2 is a special case of Algorithm 3 which is in turn special case of Algorithm 5. Similarly, Theorem 2 is a special case of Theorem 5.1 which is again special case of Theorem C.6.*

## C.6   LOCAL STOCHASTIC ALGORITHMS

In this section, we present two more algorithms – Local SGD with partial variance reduction (Algorithm 7) and Local SGD without variance reduction (Algorithm 6). While Algorithm 6 uses no control variates at all (thus is essentially Algorithm 1 where local gradient descent steps are replaced with local SGD steps), Algorithm 7 constructs control variates for $\psi$ only, resulting in locally drifted SGD algorithm (with the constant drift between each consecutive rounds of communication). While we do not present the convergence rates of the methods here, we shall notice they can be easily obtained using the framework from (Gorbunov et al., 2020).

---

**Algorithm 6** Loopless Local SGD (L2SGD)

---

**Input:** $x_1^0 = \cdots = x_n^0 \in \mathbb{R}^d$, stepsize $\alpha$, probability $p$
**for** $k = 0, 1, \ldots$ **do**
    $\xi = 1$ with probability $p$ and $0$ with probability $1 - p$
    **if** $\xi = 0$ **then**
        All Devices $i = 1, \ldots, n$:
            Sample $j \in \{1, \ldots, m\}$ (uniformly at random)
            $g_i^k = \frac{1}{n(1-p)} \left( \nabla f_{i,j}'(x_i^k) \right)$
            $x_i^{k+1} = x_i^k - \alpha g_i^k$
    **else**
        Master computes the average $\bar{x}^k = \frac{1}{n} \sum_{i=1}^n x_i^k$
        Master does for all $i = 1, \ldots, n$:
            $g_i^k = \frac{\lambda}{np}(x_i^k - \bar{x}^k)$
            Set $x_i^{k+1} = x_i^k - \alpha g_i^k$
    **end if**
**end for**

---

**Algorithm 7** Loopless Local SGD with partial variance reduction (L2SGD2)

---

**Input:** $x_1^0 = \cdots = x_n^0 \in \mathbb{R}^d$, stepsize $\alpha$, probability $p$
$\boldsymbol{\Psi}_i^0 = 0 \in \mathbb{R}^d$ (for $i = 1, \ldots, n$)
**for** $k = 0, 1, \ldots$ **do**
    $\xi = 1$ with probability $p$ and $0$ with probability $1 - p$
    **if** $\xi = 0$ **then**
        All Devices $i = 1, \ldots, n$:
            Sample $j \in \{1, \ldots, m\}$ (uniformly at random)
            $g_i^k = \frac{1}{n(1-p)} \left( \nabla f_{i,j}'(x_i^k) \right) + \frac{1}{n} \boldsymbol{\Psi}_i^k$
            $x_i^{k+1} = x_i^k - \alpha g_i^k$
            Set $\boldsymbol{\Psi}_i^{k+1} = \boldsymbol{\Psi}_i^k$
    **else**
        Master computes the average $\bar{x}^k = \frac{1}{n} \sum_{i=1}^n x_i^k$
        Master does for all $i = 1, \ldots, n$:
            $g_i^k = \frac{\lambda}{np}(x_i^k - \bar{x}^k) - \frac{p^{-1}-1}{n} \boldsymbol{\Psi}_i^k$
            Set $x_i^{k+1} = x_i^k - \alpha g_i^k$
            Set $\boldsymbol{\Psi}_i^{k+1} = \lambda(x_i^k - \bar{x}^k)$
    **end if**
**end for**

---

# D    MISSING LEMMAS AND PROOFS

## D.1    GRADIENT AND HESSIAN OF $\psi$

**Lemma D.1** *Let $\mathbf{I}$ be the $d \times d$ identity matrix and $\mathbf{I}_n$ be $n \times n$ identity matrix. Then, we have*

$$\nabla^2 \psi(x) = \tfrac{1}{n} \left( \mathbf{I}_n - \tfrac{1}{n} ee^\top \right) \otimes \mathbf{I} \quad \text{and} \quad \nabla \psi(x) = \tfrac{1}{n} \left( x - \begin{pmatrix} \bar{x} \\ \vdots \\ \bar{x} \\ \bar{x} \\ \bar{x} \\ \vdots \\ \bar{x} \end{pmatrix} \right).$$

*Furthermore, $L_\psi = \tfrac{1}{n}$.*

**Proof:**

Let $\mathbf{O}$ the $d \times d$ zero matrix and let

$$\mathbf{Q}_i := [\underbrace{\mathbf{O}, \dots, \mathbf{O}}_{i-1}, \mathbf{I}, \underbrace{\mathbf{O}, \dots, \mathbf{O}}_{n-i}] \in \mathbb{R}^{d \times dn}$$

and $\mathbf{Q} := [\mathbf{I}, \dots, \mathbf{I}] \in \mathbb{R}^{d \times dn}$. Note that $x_i = \mathbf{Q}_i x$, and $\bar{x} = \tfrac{1}{n} \mathbf{Q} x$. So,

$$\psi(x) = \tfrac{1}{2n} \sum_{i=1}^n \left\| \mathbf{Q}_i x - \tfrac{1}{n} \mathbf{Q} x \right\|^2 = \tfrac{1}{2n} \sum_{i=1}^n \left\| \left( \mathbf{Q}_i - \tfrac{1}{n} \mathbf{Q} \right) x \right\|^2.$$

The Hessian of $\psi$ is

$$\begin{aligned}
\nabla^2 \psi(x) &= \tfrac{1}{n} \sum_{i=1}^n \left( \mathbf{Q}_i - \tfrac{1}{n} \mathbf{Q} \right)^\top \left( \mathbf{Q}_i - \tfrac{1}{n} \mathbf{Q} \right) \\
&= \tfrac{1}{n} \sum_{i=1}^n \left( \mathbf{Q}_i^\top \mathbf{Q}_i - \tfrac{1}{n} \mathbf{Q}_i^\top \mathbf{Q} - \tfrac{1}{n} \mathbf{Q}^\top \mathbf{Q}_i + \tfrac{1}{n^2} \mathbf{Q}^\top \mathbf{Q} \right) \\
&= \tfrac{1}{n} \sum_{i=1}^n \mathbf{Q}_i^\top \mathbf{Q}_i - \tfrac{1}{n} \sum_{i=1}^n \tfrac{1}{n} \mathbf{Q}_i^\top \mathbf{Q} - \tfrac{1}{n} \sum_{i=1}^n \tfrac{1}{n} \mathbf{Q}^\top \mathbf{Q}_i + \tfrac{1}{n} \sum_{i=1}^n \tfrac{1}{n^2} \mathbf{Q}^\top \mathbf{Q} \\
&= \tfrac{1}{n} \sum_{i=1}^n \mathbf{Q}_i^\top \mathbf{Q}_i - \tfrac{1}{n^2} \mathbf{Q}^\top \mathbf{Q}
\end{aligned}$$

and by plugging in for $\mathbf{Q}$ and $\mathbf{Q}_i$, we get

$$\begin{aligned}
\nabla^2 \psi(x) &= \tfrac{1}{n} \begin{pmatrix}
\left(1 - \tfrac{1}{n}\right) \mathbf{I} & -\tfrac{1}{n}\mathbf{I} & -\tfrac{1}{n}\mathbf{I} & \cdots & -\tfrac{1}{n}\mathbf{I} \\
-\tfrac{1}{n}\mathbf{I} & \left(1 - \tfrac{1}{n}\right) \mathbf{I} & -\tfrac{1}{n}\mathbf{I} & \cdots & -\tfrac{1}{n}\mathbf{I} \\
-\tfrac{1}{n}\mathbf{I} & -\tfrac{1}{n}\mathbf{I} & \left(1 - \tfrac{1}{n}\right) \mathbf{I} & \cdots & -\tfrac{1}{n}\mathbf{I} \\
\vdots & \vdots & \vdots & & \vdots \\
-\tfrac{1}{n}\mathbf{I} & -\tfrac{1}{n}\mathbf{I} & -\tfrac{1}{n}\mathbf{I} & \cdots & \left(1 - \tfrac{1}{n}\right) \mathbf{I}
\end{pmatrix} \\
&= \tfrac{1}{n} \begin{pmatrix}
\left(1 - \tfrac{1}{n}\right) & -\tfrac{1}{n} & -\tfrac{1}{n} & \cdots & -\tfrac{1}{n} \\
-\tfrac{1}{n} & \left(1 - \tfrac{1}{n}\right) & -\tfrac{1}{n} & \cdots & -\tfrac{1}{n} \\
-\tfrac{1}{n} & -\tfrac{1}{n} & \left(1 - \tfrac{1}{n}\right) & \cdots & -\tfrac{1}{n} \\
\vdots & \vdots & \vdots & & \vdots \\
-\tfrac{1}{n} & -\tfrac{1}{n} & -\tfrac{1}{n} & \cdots & \left(1 - \tfrac{1}{n}\right)
\end{pmatrix} \otimes \mathbf{I} \\
&= \tfrac{1}{n} \left( \mathbf{I}_n - \tfrac{1}{n} ee^\top \right) \otimes \mathbf{I}.
\end{aligned}$$

Notice that $\mathbf{I}_n - \tfrac{1}{n} ee^\top$ is a circulant matrix, with eigenvalues 1 (multiplicity $n-1$) and 0 (multiplicity 1). Since the eigenvalues of a Kronecker product of two matrices are the products of pairs of eigenvalues of the these matrices, we have

$$\lambda_{\max}(\nabla^2 \psi(x)) = \lambda_{\max} \left( \tfrac{1}{n} \left( \mathbf{I}_n - \tfrac{1}{n} ee^\top \right) \otimes \mathbf{I} \right) = \tfrac{1}{n} \lambda_{\max} \left( \mathbf{I}_n - \tfrac{1}{n} ee^\top \right) = \tfrac{1}{n}.$$

So, $L_\psi = \frac{1}{n}$.

The gradient of $\psi$ is given by

$$
\begin{aligned}
\nabla\psi(x) &= \frac{1}{n}\sum_{i=1}^{n}\left(\mathbf{Q}_i - \tfrac{1}{n}\mathbf{Q}\right)^\top\left(\mathbf{Q}_i - \tfrac{1}{n}\mathbf{Q}\right)x \\[2mm]
&= \frac{1}{n}\sum_{i=1}^{n}\left(\mathbf{Q}_i^\top\mathbf{Q}_i - \tfrac{1}{n}\mathbf{Q}_i^\top\mathbf{Q} - \tfrac{1}{n}\mathbf{Q}^\top\mathbf{Q}_i + \tfrac{1}{n^2}\mathbf{Q}^\top\mathbf{Q}\right)x \\[2mm]
&= \frac{1}{n}\sum_{i=1}^{n}\left[\begin{pmatrix}0\\\vdots\\0\\x_i\\0\\\vdots\\0\end{pmatrix} - \begin{pmatrix}0\\\vdots\\0\\\bar{x}\\0\\\vdots\\0\end{pmatrix} - \begin{pmatrix}x_i/n\\\vdots\\x_i/n\\x_i/n\\x_i/n\\\vdots\\x_i/n\end{pmatrix} + \begin{pmatrix}\bar{x}/n\\\vdots\\\bar{x}/n\\\bar{x}/n\\\bar{x}/n\\\vdots\\\bar{x}/n\end{pmatrix}\right] \\[2mm]
&= \frac{1}{n}\left(\sum_{i=1}^{n}\begin{pmatrix}0\\\vdots\\0\\x_i\\0\\\vdots\\0\end{pmatrix} - \sum_{i=1}^{n}\begin{pmatrix}0\\\vdots\\0\\\bar{x}\\0\\\vdots\\0\end{pmatrix} - \sum_{i=1}^{n}\begin{pmatrix}x_i/n\\\vdots\\x_i/n\\x_i/n\\x_i/n\\\vdots\\x_i/n\end{pmatrix} + \sum_{i=1}^{n}\begin{pmatrix}\bar{x}/n\\\vdots\\\bar{x}/n\\\bar{x}/n\\\bar{x}/n\\\vdots\\\bar{x}/n\end{pmatrix}\right) \\[2mm]
&= \frac{1}{n}\left(x - \begin{pmatrix}\bar{x}\\\vdots\\\bar{x}\\\bar{x}\\\bar{x}\\\vdots\\\bar{x}\end{pmatrix} - \begin{pmatrix}\bar{x}\\\vdots\\\bar{x}\\\bar{x}\\\bar{x}\\\vdots\\\bar{x}\end{pmatrix} + \begin{pmatrix}\bar{x}\\\vdots\\\bar{x}\\\bar{x}\\\bar{x}\\\vdots\\\bar{x}\end{pmatrix}\right) \\[2mm]
&= \frac{1}{n}\left(x - \begin{pmatrix}\bar{x}\\\vdots\\\bar{x}\\\bar{x}\\\bar{x}\\\vdots\\\bar{x}\end{pmatrix}\right).
\end{aligned}
$$

## D.2 Proof of Theorem 3.1

For any $\lambda, \theta \geq 0$ we have

$$
\begin{aligned}
f(x(\lambda)) + \lambda\psi(x(\lambda)) &\leq f(x(\theta)) + \lambda\psi(x(\theta)) && (15) \\
f(x(\theta)) + \theta\psi(x(\theta)) &\leq f(x(\lambda)) + \theta\psi(x(\lambda)). && (16)
\end{aligned}
$$

By adding inequalities (15) and (16), we get

$$
(\theta - \lambda)(\psi(x(\lambda)) - \psi(x(\theta))) \geq 0,
$$

which means that $\psi(x(\lambda))$ is decreasing in $\lambda$. Assume $\lambda \geq \theta$. From the (16) we get

$$
f(x(\lambda)) \geq f(x(\theta)) + \theta(\psi(x(\theta)) - \psi(x(\lambda))) \geq f(x(\theta)),
$$

where the last inequality follows since $\theta \geq 0$ and since $\psi(x(\theta)) \geq \psi(x(\lambda))$. So, $f(x(\lambda))$ is increasing.

Notice that since $\psi$ is a non-negative function and since $x(\lambda)$ minimizes $F$ and $\psi(x(\infty)) = 0$, we have

$$f(x(0)) \leq f(x(\lambda)) \leq f(x(\lambda)) + \lambda\psi(x(\lambda)) \leq f(x(\infty)),$$

which implies (3) and (4).

### D.3  PROOF OF THEOREM 3.2

The equation $\nabla F(x(\lambda)) = 0$ can be equivalently written as

$$\nabla f_i(x_i(\lambda)) + \lambda(x_i(\lambda) - \overline{x}(\lambda)) = 0, \qquad i = 1, 2, \ldots, n,$$

which is identical to (5). Averaging these identities over $i$, we get

$$\overline{x}(\lambda) = \overline{x}(\lambda) - \frac{1}{\lambda}\frac{1}{n}\sum_{i=1}^{n}\nabla f_i(x_i(\lambda)),$$

which implies

$$\sum_{i=1}^{n}\nabla f_i(x_i(\lambda)) = 0.$$

Further, we have

$$\psi(x(\lambda)) = \frac{1}{2n}\sum_{i=1}^{n}\|x_i(\lambda) - \overline{x}(\lambda)\|^2 = \frac{1}{2n\lambda^2}\sum_{i=1}^{n}\|\nabla f_i(x_i(\lambda))\|^2 = \frac{1}{2\lambda^2}\|\nabla f(x(\lambda))\|^2,$$

as desired.

### D.4  PROOF OF THEOREM 3.3

First, observe that

$$\|\nabla P(\overline{x}(\lambda))\|^2 = \left\|\frac{1}{n}\sum_i\nabla f_i(\overline{x}(\lambda))\right\|^2 = \left\|\frac{1}{n}\sum_i\nabla f_i(\overline{x}(\lambda)) - \frac{1}{n}\sum_i\nabla f_i(x_i(\lambda))\right\|^2,$$

where the second identity is due to Theorem 3.2 which says that $\frac{1}{n}\sum_i\nabla f_i(x_i(\lambda)) = 0$. By applying Jensen's inequality and Lipschitz continuity of functions $f_i$, we get

$$\|\nabla P(\overline{x}(\lambda))\|^2 \leq \frac{1}{n}\sum_i\|\nabla f_i(\overline{x}(\lambda)) - \nabla f_i(x_i(\lambda))\|^2 \leq \frac{L^2}{n}\sum_i\|\overline{x}(\lambda) - x_i(\lambda)\|^2 = 2L^2\psi(x(\lambda)).$$

It remains to apply (3) and notice that $P$ is strongly convex and thus $x(\infty)$ is indeed the unique minimizer.

### D.5  PROOF OF THEOREM 4.2

We first show that our gradient estimator $G(x)$ satisfies the expected smoothness property (Gower et al., 2018; 2019).

**Lemma D.2** *Let* $\mathcal{L} := \frac{1}{n}\max\left\{\frac{L}{1-p}, \frac{\lambda}{p}\right\}$ *and* $\sigma^2 := \frac{1}{n^2}\sum_{i=1}^{n}\left(\frac{1}{1-p}\|\nabla f_i(x_i(\lambda))\|^2 + \frac{\lambda^2}{p}\|x_i(\lambda) - \overline{x}(\lambda)\|^2\right)$. *Then for all* $x \in \mathbb{R}^d$ *we have the inequalities* $\mathbb{E}\left[\|G(x) - G(x(\lambda))\|^2\right] \leq 2\mathcal{L}\left(F(x) - F(x(\lambda))\right)$ *and*

$$\mathbb{E}\left[\|G(x)\|^2\right] \leq 4\mathcal{L}(F(x) - F(x(\lambda))) + 2\sigma^2.$$

Next, Theorem 4.2 from Lemma D.2 by applying Theorem 3.1 from (Gower et al., 2019).

## D.6   Proof of Lemma D.2

We first have

$$
\begin{aligned}
\mathbb{E}\left[\|G(x) - G(x(\lambda))\|^2\right] &= (1-p)\left\|\frac{\nabla f(x)}{1-p} - \frac{\nabla f(x(\lambda))}{1-p}\right\|^2 + p\left\|\lambda\frac{\nabla\psi(x)}{p} - \lambda\frac{\nabla\psi(x(\lambda))}{p}\right\|^2 \\
&= \frac{1}{1-p}\|\nabla f(x) - \nabla f(x(\lambda))\|^2 + \frac{\lambda^2}{p}\|\nabla\psi(x) - \nabla\psi(x(\lambda))\|^2 \\
&\leq \frac{2L_f}{1-p}D_f(x, x(\lambda)) + \frac{2\lambda^2 L_\psi}{p}D_\psi(x, x(\lambda)) \\
&= \frac{2L}{n(1-p)}D_f(x, x(\lambda)) + \frac{2\lambda^2}{np}D_\psi(x, x(\lambda)).
\end{aligned}
$$

Since $D_f + \lambda D_\psi = D_F$ and $\nabla F(x(\lambda)) = 0$, we can continue:

$$
\begin{aligned}
\mathbb{E}\left[\|G(x) - G(x(\lambda))\|^2\right] &\leq \frac{2}{n}\max\left\{\frac{L}{1-p}, \frac{\lambda}{p}\right\}D_F(x, x(\lambda)) \\
&= \frac{2}{n}\max\left\{\frac{L}{1-p}, \frac{\lambda}{p}\right\}(F(x) - F(x(\lambda))).
\end{aligned}
$$

Next, note that

$$
\begin{aligned}
\sigma^2 &= \frac{1}{n^2}\sum_{i=1}^{n}\left(\frac{1}{1-p}\|\nabla f_i(x_i(\lambda))\|^2 + \frac{\lambda^2}{p}\|x_i(\lambda) - \overline{x}(\lambda)\|^2\right) \\
&= \frac{1}{1-p}\|\nabla f(x(\lambda))\|^2 + \frac{\lambda^2}{p}\|\nabla\psi(x(\lambda))\|^2 \\
&= (1-p)\left\|\frac{\nabla f(x(\lambda))}{1-p}\right\|^2 + p\left\|\frac{\lambda\nabla\psi(x(\lambda))}{p}\right\|^2 \\
&= \mathbb{E}\left[\|G(x(\lambda))\|^2\right].
\end{aligned}
$$

Therefore, we have

$$
\begin{aligned}
\mathbb{E}\left[\|G(x)\|^2\right] &\leq \mathbb{E}\left[\|G(x) - G(x(\lambda))\|^2\right] + 2\mathbb{E}\left[\|G(x(\lambda))\|^2\right] \\
&\overset{\text{Lemma }D.2+(17)}{\leq} 4\mathcal{L}(F(x) - F(x(\lambda))) + 2\sigma^2,
\end{aligned}
$$

as desired.

## D.7   Proof of Corollary 4.3

Firstly, to minimize the total number of iterations, it suffices to minimize $\mathcal{L}$ which is achieved with $p^\star = \frac{\lambda}{L+\lambda}$. Let us look at the communication. Fix $\varepsilon > 0$, choose $\alpha = \frac{1}{2\mathcal{L}}$ and let $k = \frac{2n\mathcal{L}}{\mu}\log\frac{1}{\varepsilon}$, so that

$$
\left(1 - \frac{\mu}{2n\mathcal{L}}\right)^k \leq \varepsilon.
$$

The expected number of communications to achieve this goal is equal to

$$
\begin{aligned}
\mathrm{Comm}_p &:= p(1-p)k \\
&= p(1-p)\frac{2\max\left\{\frac{L}{1-p}, \frac{\lambda}{p}\right\}}{\mu}\log\frac{1}{\varepsilon} \\
&= \frac{2\max\{pL, (1-p)\lambda\}}{\mu}\log\frac{1}{\varepsilon}.
\end{aligned}
$$

The quantity $\mathrm{Comm}_p$ is minimized by choosing any $p$ such that $pL = (1-p)\lambda$, i.e., for $p = \frac{\lambda}{\lambda+L} = p^\star$, as desired. The optimal expected number of communications is therefore equal to

$$
\mathrm{Comm}_{p^\star} = \frac{2\lambda}{\lambda+L}\frac{L}{\mu}\log\frac{1}{\varepsilon}.
$$

## D.8  PROOF OF COROLLARY 5.2

Firstly, to minimize the total number of iterations, it suffices to solve

$$\min \max \left\{ \tfrac{4L'+\mu m}{(1-p)\mu}, \tfrac{4\lambda+\mu}{p\mu} \right\},$$

which is achieved with $p = p^\star = \frac{4\lambda+\mu}{4L'+4\lambda+(m+1)\mu}$.

The expected number of communications to reach $\varepsilon$-solution is

$$
\begin{aligned}
\mathrm{Comm}_p &= p(1-p) \max \left\{ \tfrac{4L'+\mu m}{(1-p)\mu}, \tfrac{4\lambda+\mu}{p\mu} \right\} \log \tfrac{1}{\varepsilon} \\
&= \tfrac{\max\left\{ p(4L'+\mu m),(1-p)(4\lambda+\mu) \right\}}{\mu} \log \tfrac{1}{\varepsilon}.
\end{aligned}
$$

Minimizing the above in $p$ yield $p = p^\star = \frac{4\lambda+\mu}{4L'+4\lambda+(m+1)\mu}$, as desired. The optimal expected number of communications is therefore equal to

$$\mathrm{Comm}_{p^\star} = \tfrac{4\lambda+\mu}{4L'+4\lambda+(m+1)\mu} \left( 4\tfrac{L'}{\mu} + m \right) \log \tfrac{1}{\varepsilon}.$$

## D.9  PROOF OF THEOREMS 5.1, C.6, AND C.7

Note first that Algorithm 3 is a special case of Algorithm 5, and Theorem 5.1 immediately follows from Theorem C.6. Therefore it suffices to show Theorems C.6, and C.7. In order to do so, we will cast Algorithm 5 as a special case of GJS from (Hanzely & Richtárik, 2019). As a consequence, Theorem C.6 will be a special cases of Theorem 5.2 from (Hanzely & Richtárik, 2019).

### D.9.1  GJS

In this section, we quickly summarize results from (Hanzely & Richtárik, 2019), which we cast to sho convergence rate of Algorithm 3. GJS (Hanzely & Richtárik, 2019) is a method to solve regularized empirical risk minimization objective, i.e.,

$$\min_{x\in\mathbb{R}^d} \tfrac{1}{n} \sum_{j=1}^{n} f_j(x) + R(x). \tag{17}$$

Defining $\mathbf{G}(x) \coloneqq [\nabla f_1(x), \ldots, \nabla f_n(x)]$, we observe $\mathcal{S}\mathbf{G}(x), \mathcal{U}\mathbf{G}(x)$ every iteration where $\mathcal{S}$ is random linear projection operator and $\mathcal{U}$ is random linear operator which is identity on expectation. Based on this random gradient information, GJS (Algorithm 8) constructs variance reduced gradient estimator $g$ and takes a proximal step in that direction.

---

**Algorithm 8** Generalized `JacSketch` (GJS) (Hanzely & Richtárik, 2019)

---
1: **Parameters:** Stepsize $\alpha > 0$, random projector $\mathcal{S}$ and unbiased sketch $\mathcal{U}$
2: **Initialization:** Choose solution estimate $x^0 \in \mathbb{R}^d$ and Jacobian estimate $\mathbf{J}^0 \in \mathbb{R}^{d\times n}$
3: **for** $k = 0, 1, \ldots$ **do**
4:    Sample realizations of $\mathcal{S}$ and $\mathcal{U}$, and perform sketches $\mathcal{S}\mathbf{G}(x^k)$ and $\mathcal{U}\mathbf{G}(x^k)$
5:    $\mathbf{J}^{k+1} = \mathbf{J}^k - \mathcal{S}(\mathbf{J}^k - \mathbf{G}(x^k))$ $\qquad\qquad\qquad$ update the Jacobian estimate
6:    $g^k = \tfrac{1}{n}\mathbf{J}^k e + \tfrac{1}{n}\mathcal{U}\left(\mathbf{G}(x^k) - \mathbf{J}^k\right)e$ $\qquad\qquad$ construct the gradient estimator
7:    $x^{k+1} = \mathrm{prox}_{\alpha R}(x^k - \alpha g^k)$ $\qquad\qquad\qquad$ perform the proximal `SGD` step
8: **end for**

---

Next we quickly summarize theory of GJS.

**Assumption D.1** *Problem* (17) *has a unique minimizer $x^\star$, and $f$ is $\mu$-quasi strongly convex, i.e.,*

$$f(x^\star) \geq f(y) + \langle \nabla f(y), x^\star - y \rangle + \tfrac{\mu}{2} \|y - x^\star\|^2, \quad \forall y \in \mathbb{R}^d, \tag{18}$$

*Functions $f_j$ are convex and $\mathbf{M}_j$-smooth for some $\mathbf{M}_j \succeq 0$, i.e.,*

$$f_j(y) + \langle \nabla f_j(y), x - y \rangle \leq f_j(x) \leq f_j(y) + \langle \nabla f_j(y), x - y \rangle + \tfrac{1}{2}\|y - x\|_{\mathbf{M}_j}^2, \quad \forall x, y \in \mathbb{R}^d. \tag{19}$$

**Theorem D.3 (Slight simplification of Theorem 5.2 from (Hanzely & Richtárik, 2019))** *Let Assumption D.1 hold. Define* $\mathcal{M}(\mathbf{X}) := [\mathbf{M}_1 X_{:,1}, \dots, \mathbf{M}_n X_{:,n}]$ *Let* $\mathcal{B}$ *be any linear operator commuting with* $\mathcal{S}$, *and assume* $\mathcal{M}^{\dagger\frac{1}{2}}$ *commutes with* $\mathcal{S}$. *Define the Lyapunov function*

$$\Psi^k \quad := \quad \left\| x^k - x^\star \right\|^2 + \alpha \left\| \mathcal{B} \mathcal{M}^{\dagger\frac{1}{2}} \left( \mathbf{J}^k - \mathbf{G}(x^\star) \right) \right\|^2, \tag{20}$$

*where* $\{x^k\}$ *and* $\{\mathbf{J}^k\}$ *are the random iterates produced by Algorithm 8 with stepsize* $\alpha > 0$. *Suppose that* $\alpha$ *and* $\mathcal{B}$ *are chosen so that*

$$\frac{2\alpha}{n^2} \mathbb{E}\left[ \|\mathcal{U}\mathbf{X}e\|^2 \right] + \left\| (\mathcal{I} - \mathbb{E}[\mathcal{S}])^{\frac{1}{2}} \mathcal{B} \mathcal{M}^{\dagger\frac{1}{2}} \mathbf{X} \right\|^2 \quad \leq \quad (1 - \alpha\mu) \left\| \mathcal{B} \mathcal{M}^{\dagger\frac{1}{2}} \mathbf{X} \right\|^2 \tag{21}$$

*and*

$$\frac{2\alpha}{n^2} \mathbb{E}\left[ \|\mathcal{U}\mathbf{X}e\|^2 \right] + \left\| (\mathbb{E}[\mathcal{S}])^{\frac{1}{2}} \mathcal{B} \mathcal{M}^{\dagger\frac{1}{2}} \mathbf{X} \right\|^2 \quad \leq \quad \frac{1}{n} \left\| \mathcal{M}^{\dagger\frac{1}{2}} \mathbf{X} \right\|^2. \tag{22}$$

*for all* $\mathbf{X} \in \mathbb{R}^{d\times n}$. *Then for all* $k \geq 0$, *we have* $\mathbb{E}\left[\Psi^k\right] \leq (1-\alpha\mu)^k \Psi^0$.

### D.9.2 VARIANCE REDUCED LOCAL SGD AS SPECIAL CASE OF GJS

Let $\Omega(i,j) := j + \sum_{l=1}^{i-1} m_i$ In order to case problem (12) as a special case of 17, denote $n := N+1$, $f_{\Omega(i,j)}(x) := \frac{N+1}{N} f'_{i,j}(x_i)$ and $f_n := (N+1)\psi$. Therefore the objective (12) becomes

$$\min_{x \in \mathbb{R}^{Nd}} \Upsilon(x) := \frac{1}{n} \sum_{j=1}^{n} f_j(x) + R(x). \tag{23}$$

Let $v \in \mathbb{R}^{n-1}$ be such that $v_{\Omega(i,j)} = \frac{N+1}{N} v_{i,j}$ and as a consequence of (14) we have

$$\mathbb{E}\left[ \left\| \sum_{j\in S_i} \mathbf{M}_{i,j}^{\frac{1}{2}} h_{i,j} \right\|^2 \right] \leq \sum_{j=1}^{m_i} p_{i,j} v_{\Omega(i,j)} \|h_{i,j}\|^2, \qquad \forall\, 1 \leq i \leq n,\ \forall h_{i,j} \in \mathbb{R}^d, j \in \{1,\dots,m_i\}. \tag{24}$$

At the same time, $\Upsilon$ is $\mu := \frac{\mu}{n}$ strongly convex.

### D.9.3 PROOF OF THEOREM C.6 AND THEOREM C.7

Let $e \in \mathbb{R}^d$ be a vector of ones and $p^i \in \mathbb{R}^N$ is such that $p^i_j = p_{i,j}$ if $j \in \{1,\dots,m_i\}$, otherwise $p^i_j = 0$. Given the notation, random operator $\mathcal{U}$ is chosen as

$$\mathcal{U}\mathbf{X} = \begin{cases} (1-p)^{-1} \sum_{i=1}^{n} \left( p_i^{-1} e \left( (p^i)^{-1} \right)^\top \right) \circ \left( \mathbf{X}_{:m_i} \left( \sum_{j\in S_i} e_j e_j^\top \right) \right) & \text{w.p.} \quad (1-p) \\ p^{-1} \mathbf{X}_{:,n} & \text{w.p.} \quad p \end{cases}$$

We next give two options on how to update Jacobian – first one is SAGA-like, second one is SVRG like.

SAGA-like: $\quad (\mathcal{S}\mathbf{X})_{:,m_i} = \begin{cases} \mathbf{X}_{:,S_i} = \mathbf{X}_{:m_i} \left( \sum_{j\in S_i} e_j e_j^\top \right), & \text{w.p.} \quad (1-p)p_i, \\ 0 & \text{w.p.} \quad (1-p)(1-p_i)+p \end{cases}$

$\quad\quad\quad\quad\quad (\mathcal{S}\mathbf{X})_{:,n} = \begin{cases} \mathbf{X}_{:,n} & \text{w.p.} \quad p \\ 0 & \text{w.p.} \quad 1-p \end{cases}$

SVRG-like: $\quad (\mathcal{S}\mathbf{X})_{:,m_i} = \begin{cases} \mathbf{X}_{:m_i} b_i;\ b_i = \begin{cases} 1 & \text{w.p.} \quad p_i \\ 0 & \text{w.p.} \quad 1-p_i \end{cases} & \text{w.p.} \quad (1-p)p_i \\ 0 & \text{w.p.} \quad (1-p)(1-p_i)+p \end{cases}$

$\quad\quad\quad\quad\quad (\mathcal{S}\mathbf{X})_{:,n} = \begin{cases} \mathbf{X}_{:,n} & \text{w.p.} \quad p \\ 0 & \text{w.p.} \quad 1-p. \end{cases}$

We can now proceed with the proof of Theorem C.6 and Theorem C.7. As $\nabla f_i(x) - \nabla f_i(y) \in$ Range $(\mathbf{M}_i)$, we must have

$$\mathbf{G}(x^k) - \mathbf{G}(x^\star) = \mathcal{M}^\dagger \mathcal{M} \left( \mathbf{G}(x^k) - \mathbf{G}(x^\star) \right) \tag{25}$$

and

$$\mathbf{J}^k - \mathbf{G}(x^\star) = \mathcal{M}^\dagger \mathcal{M} \left( \mathbf{J}^k - \mathbf{G}(x^\star) \right). \tag{26}$$

Due to (26), (25), inequalities (21) and (22) with choice $\mathbf{Y} = \mathcal{M}^{\dagger \frac{1}{2}} \mathbf{X}$ become respectively:

$$\frac{2\alpha}{n^2} p^{-1} \|\mathbf{M}_n^{\frac{1}{2}} \mathbf{Y}_{:,n}\|^2 + \frac{2\alpha^2}{n^2}(1-p)^{-1} \sum_{i=1}^n \mathbb{E} \left[ \left\| p_i^{-1} \sum_{j \in S_i} p_{i,j}^{-1} \mathbf{M}_{i,j}^{\frac{1}{2}} \mathbf{Y}_{:j} \right\|^2 \right] + \left\| (\mathcal{I} - \mathbb{E}\left[\mathcal{S}\right])^{\frac{1}{2}} \mathcal{B}(\mathbf{Y}) \right\|^2$$

$$\leq (1 - \alpha\mu) \|\mathcal{B}(\mathbf{Y})\|^2 \tag{27}$$

$$\frac{2\alpha}{n^2} p^{-1} \|\mathbf{M}_n^{\frac{1}{2}} \mathbf{Y}_{:,n}\|^2 + \frac{2\alpha^2}{n^2}(1-p)^{-1} \sum_{i=1}^n \mathbb{E} \left[ \left\| p_i^{-1} \sum_{j \in S_i} p_{i,j}^{-1} \mathbf{M}_{i,j}^{\frac{1}{2}} \mathbf{Y}_{:j} \right\|^2 \right] + \left\| (\mathbb{E}\left[\mathcal{S}\right])^{\frac{1}{2}} \mathcal{B}(\mathbf{Y}) \right\|^2 \leq \frac{1}{n} \|\mathbf{Y}\|^2 \tag{28}$$

Above, we have used

$$E\|\mathcal{U}\mathbf{X}e\|^2 = \mathbb{E} \left[ \|\mathcal{U}\mathcal{M}^{\frac{1}{2}} \mathbf{Y}e\|^2 \right] = p^{-1} \|\mathbf{M}_n^{\frac{1}{2}} \mathbf{Y}_{:,n}\|^2 + (1-p)^{-1} \sum_{i=1}^n \mathbb{E} \left[ \left\| p_i^{-1} \sum_{j \in S_i} p_{i,j}^{-1} \mathbf{M}_{i,j}^{\frac{1}{2}} \mathbf{Y}_{:j} \right\|^2 \right].$$

Note that $\mathbb{E}\left[\mathcal{S}(\mathbf{X})\right] = \mathbf{X} \cdot \mathrm{Diag}\left((1-p)(p \circ p), p\right)$ where $p \in \mathbb{R}^{n-1}$ such that $p_{\Omega(i,j)} = p_{i,j}$. Using (24), setting $\mathcal{B}$ to be right multiplication with $\mathrm{Diag}(b)$ and noticing that $\lambda_{\max}\mathbf{M}_n = n\lambda$ it suffices to have

$$\frac{2\alpha}{n} p^{-1} \lambda + (1-p) b_n^2 \leq (1 - \alpha\mu) b_n^2$$
$$\frac{2\alpha}{n^2}(1-p)^{-1} p_{i,j}^{-1} p_i^{-1} v_{\Omega(i,j)} + (1 - (1-p)p_{i,j}p_i) b_j^2 \leq (1 - \alpha\mu) b_j^2 \qquad \forall j \in \{1, \ldots, m_i\}, i \leq n$$

$$\frac{2\alpha}{n} p^{-1} \lambda + p b_n^2 \leq \frac{1}{n}$$
$$\frac{2\alpha}{n^2}(1-p)^{-1} p_{i,j}^{-1} p_i^{-1} v_{\Omega(i,j)} + (1-p)p_{i,j}p_i b_j^2 \leq \frac{1}{n} \qquad \forall j \in \{1, \ldots, m_i\}, i \leq n$$

for SAGA case and

$$\frac{2\alpha}{n} p^{-1} \lambda + (1-p) b_n^2 \leq (1 - \alpha\mu) b_n^2$$
$$\frac{2\alpha}{n^2}(1-p)^{-1} p_{i,j}^{-1} p_i^{-1} v_{\Omega(i,j)} + (1 - (1-p)p_i p_i) b_j^2 \leq (1 - \alpha\mu) b_j^2 \qquad \forall j \in \{1, \ldots, m_i\}, i \leq n$$

$$\frac{2\alpha}{n} p^{-1} \lambda + p b_n^2 \leq \frac{1}{n}$$
$$\frac{2\alpha}{n^2}(1-p)^{-1} p_{i,j}^{-1} p_i^{-1} v_{\Omega(i,j)} + (1-p)p_i p_i b_j^2 \leq \frac{1}{n} \qquad \forall j \in \{1, \ldots, m_i\}, i \leq n$$

for LSVRG case.

It remains to notice that to satisfy the SAGA case, it suffices to set $b_n^2 = \frac{1}{2np}, b_{\Omega(i,j)}^2 = \frac{1}{2n(1-p)p_{i,j}p_i}$ (for $j \in \{1, \ldots, m_i\}, i \leq n$) and $\alpha = \min \left\{ \min_{j \in \{1, \ldots, m_i\}, 1 \leq i \leq n} \frac{n(1-p)p_{i,j}p_i}{4v_{\Omega(i,j)} + n\mu}, \frac{p}{4\lambda + \mu} \right\}$.

To satisfy LSVRG case, it remains to set $b_n^2 = \frac{1}{2np}, b_{\Omega(i,j)}^2 = \frac{1}{2n(1-p)p_i p_i}$ (for $j \in \{1, \ldots, m_i\}, i \leq n$) and $\alpha = \min \left\{ \min_{j \in \{1, \ldots, m_i\}, 1 \leq i \leq n} \frac{n(1-p)p_i}{4 \frac{v_{\Omega(i,j)}}{p_{i,j}} + n\mu p_i^{-1}}, \frac{p}{4\lambda + \mu} \right\}$.

The last step to establish is to recall that $n = N + 1, v_{\Omega(i,j)} = \frac{N+1}{N} v_{i,j}$ and $\mu = \frac{\mu}{n}$ and note that the iteration complexity is $\frac{1}{\alpha\mu} \log \frac{1}{\varepsilon} = \frac{n}{\alpha\mu} \log \frac{1}{\varepsilon}$.

### D.9.4 PROOF OF THEOREM 5.1

To obtain convergence rate of Theorem 5.1, it remains to use Theorem C.6 with $p_i = 1, m_i = m$ ($\forall i \leq n$), where each machine samples (when the aggregation is not performed) individual data points with probability $\frac{1}{m}$ and thus $p_j = \frac{1}{m}$ (for all $j \leq N$). The last remaining thing is to realize that $v_j = L'$ for all $j \leq N$.

