# OpenReview forum: "Federated Learning of a Mixture of Global and Local Models"
_ICLR.cc/2021/Conference — Reject_

### Official Review · AnonReviewer1 · 2020-10-26
**Interesting insights into federated learning, possibly limited by the focus on strong convexity**

**Rating:** 6
**Confidence:** 4

**Review:**

Interesting insights into federated learning, possibly limited by the focus on strong convexity

This paper considers distributed training problems arising in the context of federated learning. It proposes a novel framing of the problem as a compromise between fitting a model locally to the data available at a device, and fitting a model globally to the data from all devices. This leads to the so-called loopless local gradient descent (L2GD) method, which is loosely related to the popular FedAvg/LocalSGD method, and also loosely related to a randomized version of the well-studied ADMM for consensus optimization problems.

The paper provides theoretical convergence guarantees for L2GD and related variance-reduced versions L2GD+ and L2GD++. As is pointed out in footnote 1, the ideas underlying the L2GD and its analysis have been previously explored in the literature. It would be worth clarifying in the paper (in footnote 1 or elsewhere) what aspects of L2GD and its analysis which are novel and not completely subsumed by the previous works Liu et al. (2017) or Wang et al. (2018). The last sentence of footnote 1 is somewhat in this direction, but is not very precise.

I generally think the perspective proposed in this paper, along with the L2GD method, are novel and interesting, and I expect they will be useful to researchers working on federated learning. The main limitation of the work, in my view, is the limited fit and interest to the broader ICLR audience. For example, the CFP emphasizes the importance of non-convex optimization, and deep learning methods and architectures for representation learning. On the other hand, this paper focuses on convex models, both for the analysis (smooth and strongly convex) and in the experiments (l2-regularized logistic regression). The analysis techniques do not appear to make use of strong global structure (beyond the definition of strong convexity). Is there reason to believe it will not be possible to provide local convergence guarantees for L2GD under more relaxed assumptions (in particular, without assuming convexity)?

A few other minor points:
* Regarding the second motivating point, see also the recent work of Woodworth et al. (arxiv:2006.04735 and arxiv:2002.07839).
* Is there any intuition why, in Fig 1, when $\lambda$ approaches $10^{1}$, the blue curve begins to decrease and orange curve begins to increase? (I.e., why the non-monotonic behaviour?)
* Minor nit.: Alg 1 seems to violate the important notion in FL that devices never reveal their private information (e.g., their local decision variables or gradients) directly to the centralized master. Rather than saying that (8) is implemented at the master, would it make more sense for each device to receive $\bar{x}^k$ from the Master and implement (8) locally? (The averaged model can be computed in a secure way using DP and secure aggregation, much the same as it is in current FL implementations.)

---

> ### Author Response · Authors · 2020-11-24
> **Reply**
>
> **Footnote 1**
>
> While the optimization problem that we study is not new on its own (we arrived at it independently and naturally by thinking about the role of local steps in FL methods), the link of such an objective with the FL is new. Specifically, the personalized FL objective (1) allows explaining the role of local steps as the local methods appear as natural applications of various SGD methods to (1). We get meaningful convergence rates, and in communication complexity, our approach shows (for the first time!) better performance than their non-local counterparts. So, we provide new links between local methods, personalization and communication complexity. This is the main novelty of our work explained as a high-level idea.
>
> **Limited fit**
>
> We decided to focus on (strongly) convex optimization in this work since the results, in that case, are easier to interpret. Until now, local methods have never been proven to outperform their non-local cousins on problems with heterogeneous data even in the convex or strongly convex case. We believe that it is important to explain properly the convex case first before moving to the non-convex one.
>
> Note that we could consider a relaxation of a strong convexity such as quasi-strong convexity or the PL condition, and we can easily derive similar rates and claim that our results apply to such a (arguably limited) non-convex case. There is nothing preventing us from deriving rates for nonconvex functions. We did not do so as there is already a lot of material covered in the paper and it does not make sense to make the coverage even more dense. We plan to do this in a follow up work, and already have some results in this direction. There is too much to be said here, and much of it is different from the strongly convex case, so indeed, mixing the convex and nonconvex cases would not be a good idea.
>
> **Minor**
>
> Re the motivation point: Thanks a lot for the reference! Indeed, we are aware of that paper, however, we decided to not include it since our work was done sooner, and hence was not influenced by this paper.
>
> **Re Fig 1**
>
> Very good question. Intuitively, the blue curve should keep increasing and the orange curve should indeed keep decreasing as we increase lambda. However, we do not have a proof of that statement. So, this does not contradict our theory nut instead provided additional computational insights complementing our theory.
>
> **Re privacy**
>
> We agree with the reviewer that our current algorithm designs did not take the privacy into the consideration. While privacy is a very important aspect of FL; in this paper, we tackle different FL challenges and thus we ignore privacy issues. However, similar to the classical local SGD methods, our algorithms can be implemented in a private fashion as well using similar ideas as you have pointed out. We will add a remark about this in the paper; thanks! Typically, papers that do theory will necessarily ignore some systems level considerations, and papers that do not ignore such things typically can't contain any meaningful theory.

---

> > ### Comment · AnonReviewer1 · 2020-11-24
> > **Acknowledging that I read your responses**
> >
> > > So, we provide new links between local methods, personalization and communication complexity. This is the main novelty of our work explained as a high-level idea.
> >
> > I recognize this contribution. I can also see, from the perspective of other reviewers, how this may seem to make the contribution appear incremental. I think it could also have been communicated more clearly (e.g., moving the contents of footnote 1 to the main body of the paper).
> >
> > > There is too much to be said here, and much of it is different from the strongly convex case, so indeed, mixing the convex and nonconvex cases would not be a good idea.
> >
> > I fully agree that your paper already covers a lot of ground, and that including more results just for the sake of covering the non-convex setting would not be a good idea. I also comment you for not stopping at restricted non-convex classes like quasi-strong or PL. It's exciting to hear that you have additional results in this direction. The intention of my comment was more to point out that results on the non-convex setting may have drawn greater enthusiasm from this crowd.
> >
> > > Indeed, we are aware of that paper, however, we decided to not include it since our work was done sooner, and hence was not influenced by this paper.
> >
> > It may be more of a limitation of the double-blind system adopted by ICLR, but unfortunately it doesn't matter whether your work was done sooner or was not influenced by those references (arxiv:2006.04735 and arxiv:2002.07839). They appeared more than a month before the ICLR submission deadline and they are very relevant to the topic of this paper, so they ought to be cited and discussed.
> >
> > > Intuitively, the blue curve should keep increasing and the orange curve should indeed keep decreasing as we increase lambda. However, we do not have a proof of that statement. So, this does not contradict our theory nut instead provided additional computational insights complementing our theory.
> >
> > Please elaborate on what you mean by "additional computational insights" here. The reply does not really address my question. You seem to agree intuitively with the point I mentioned. What happens for even larger values of $\lambda$?
> >
> > > Typically, papers that do theory will necessarily ignore some systems level considerations, and **papers that do not ignore such things typically can't contain any meaningful theory.**
> >
> > This is sad. One can always strive for a balance. Theory papers that ignore important practical considerations don't necessarily contain *meaningful* theory either.

---

> > > ### Author Response · Authors · 2020-11-24
> > > **Some more clarification, hopefully useful.**
> > >
> > > **I recognize this contribution. I can also see, from the perspective of other reviewers, how this may seem to make the contribution appear incremental. I think it could also have been communicated more clearly (e.g., moving the contents of footnote 1 to the main body of the paper).**
> > >
> > > We believe most reviewers failed to see this contribution; and that's why they made the remarks they made. Some if it is clear from the reviews. We do not believe though providing such link is incremental. It's a big shift, in our opinion, in how we believe one should view the role of local steps in local methods. The introduction of virtually all FL papers mentions that local steps are designed due to communication complexity in mind. Virtually all authors also stress the importance of non-iid assumption. But in the space of non-iid data, there are no results that suggest local methods beat their non-local counterparts in communication complexity. Our paper is the first to do so, and we view this as a substantial and not incremental contribution. And we show how degree of personalization plays an important role here.
> > >
> > > **I fully agree that your paper already covers a lot of ground, and that including more results just for the sake of covering the non-convex setting would not be a good idea. I also comment you for not stopping at restricted non-convex classes like quasi-strong or PL. It's exciting to hear that you have additional results in this direction. The intention of my comment was more to point out that results on the non-convex setting may have drawn greater enthusiasm from this crowd.**
> > >
> > > Yes, we are in agreement here. We can only hope the convex setting, and the potential for a transformation of thinking of what local steps are doing in FL optimizers that our work provides, will be appreciated as well.
> > >
> > > **It may be more of a limitation of the double-blind system adopted by ICLR, but unfortunately it doesn't matter whether your work was done sooner or was not influenced by those references (arxiv:2006.04735 and arxiv:2002.07839). They appeared more than a month before the ICLR submission deadline and they are very relevant to the topic of this paper, so they ought to be cited and discussed.**
> > >
> > > We think this is a matter of philosophy behind how science is done. We are of the belief that earlier contributions should not be required to cite later contributions, unless of course, the earlier paper evolves and changed in time. We can add these citations, as clearly they are relevant. But we would find it prudent to mention that we have read the papers only after a first complete draft of our paper was ready. This would be a fair way of handing this. Happy to do so. Readers should not be confused by inverted timelines, and timelines matter.
> > >
> > >  **Please elaborate on what you mean by "additional computational insights" here. The reply does not really address my question. You seem to agree intuitively with the point I mentioned. What happens for even larger values of  $\lambda$?**
> > >
> > > Happy to elaborate.  We do not have any theoretical statements in the paper that would dictated how the functions $A:\lambda\mapsto ||x(\lambda)-x(0)||^2$ and $B:\lambda\mapsto ||x(\lambda)-x(\infty)||^2$ should look like.
> > > Vector $x(\infty)$ is not necessarily solving the problem $$\max_{\lambda \geq 0} A(\lambda).$$ So, some amount of "oscillation" is expected for large values of $\lambda$. Perhaps this explains the kink. But we do not really know. We'll do an experiment with larger values of $\lambda$ and report on our findings. We will probably not manage to do this this by the response deadline, but we will certainly update this in our paper, whether the paper is accepted or not. It's a good suggestion.
> > >
> > > Somewhat intuitively, one should expect $A$ to be "increasing" (in the sense that it starts big and eventually becomes small) and $B$ to be "decreasing" (in the sense that is starts small and eventually becomes big). But this does not necessarily have to mean (and we do not believe it does) monotonicity in a mathematical sense.
> > >
> > > We do have statements about monotonic behavior of other quantities: $f(x(\lambda))$ and $\psi(x(\lambda))$. These influence the quantities $A$ and $B$ indirectly only.
> > >
> > > **This is sad. One can always strive for a balance. Theory papers that ignore important practical considerations don't necessarily contain meaningful theory either.**
> > >
> > > Let us be more precise as we believe there is a bit of misunderstanding here. We wanted to say the following: papers that strive for maximum applicability will contain minimal, trivial or no theory, as they need to deal with many hard practical and systems constraints that are simply beyond the reach of analytical tools. On the other hand, papers containing beautiful theory necessarily ignore many practical considerations, as only in this way can capture some isolated phenomenon analytically. Both of these extremes can be fine research. We personally have a preference for a balance.

---

> > > ### Author Response · Authors · 2020-11-25
> > > **About Figure 1**
> > >
> > > The reason why the lines in Fig 1 were non-monotonic was the numerical errors. Specifically, we did not run the algorithm that estimates $x(\lambda)$ for enough iterations when $\lambda$ is large. After re-running the code with enough precision, we can see that both blue and orange curves are monotonic; there is no kink for large values of $\lambda$. Thanks a lot for spotting this! Figure 1 is now fixed in the revised version of the manuscript.

---

### Official Review · AnonReviewer2 · 2020-10-27
**Review #2**

**Rating:** 4
**Confidence:** 5

**Review:**

## Summary

This paper proposed a new formulation of federated learning, which balances between traditional global model and purely local models. The authors discuss the advantages of the new formulation and propose a new algorithm L2GD to solve the problem. They theoretically analyzed the communication complexity of L2GD under stronly-convex settings and propose several algorithmic variants.

## Pros
1. The authors developed a set of algorithms based on L2GD and provided theoretical analysis. The efforts are appreciated.


## Cons

Unfortunately, most contributions of this paper doesn't make sense to me. I have concerns regarding novelty and correctness of the statements made by this paper. Detailed comments are listed as follows.

**New formulation**
1. In section 2, the authors claim that "we prove that the optimal local models converge to the traditional global model characterized by (1) at the rate $O(1/\lambda)$". However, I didn't find any discussions or proof around this statement in following sections. From my understanding, there should be some equations showing how $x(\lambda)-x(\infty)$ or $f(x(\lambda))-f(x(\infty))$ changes over $\lambda$. But I didn't find any.
2. In experiments, the authors didn't compare the proposed formulation with the original formulation. How much benefits one can obtain from the new formulation is unclear, making this paper to be incomplete.
3. The formulation is not new. A nearly same formulation can be found in [1]. The authors didn't notice this paper. Since [1] also proposed an algorithm EASGD to solve the new formulation, the authors are supposed to compare L2GD with the algorithm in [1].

**New algorithm: Loopless LGD**
1. The algorithm is also not new. An extremely similar algorithm has appeared in [2]. By some re-parameterization, I believe they are equivalent to each other. The authors missed this reference. They should justify the differences and compare the results.
2. It is unclear why L2GD uses random local steps instead of a fixed one. Or what are the benefits of the randomized one?

**Convergence theory**
1. The conclusions of convergence analysis are questionable. In particular, when obtaining the convergence rate, it seems that the authors completely ignore the second term in (9). In general, people use $f(x)-f(x_*)<\epsilon$ or $\|x-x_* \| < \epsilon$ to define the $\epsilon$-neighborhood of the optimum. However, in this paper, the authors use $\|x-x_* \| < \epsilon \|x_0 - x_* \| + c$ to define the neighborhood and just ignore the second term when deriving the rate. Under this definition, they draw the conclusion that L2GD can improve the communication complexity of GD. This can be misleading and questionable. In GD, we don't have the second term in (9) at all.
2. Similarly, when obtaining the best value of $p$, the authors only optimize the first term. However, the second term in (9) also depend on $p$. The authors seem to ignore it again.

**New insights about the role of local steps**
1. The authors state that "the role of local steps in gradient type methods is not to reduce communication complexity, as is generally believed. Indeed, there is no theoretical result supporting this claim in the key heterogeneous data regime." This statement is not true. It has been shown in literature (eg, [3]) that local SGD can achieve the same rate $1/\sqrt{n K}$ as synchronous SGD but only uses $O(n^{3/4} T^{3/4})$ communication rounds, while synchronous SGD uses $T$ rounds.
2. "The more local steps are taken, the more we bias the method towards the purely local models." I feel we cannot draw this conclusion from this paper's analysis. In particular, in this paper, the choice of local steps is controlled by the parameter $\lambda$ (expected local steps = $1+L/\lambda$). When we set a small $\lambda$, we get two consequences: (1) the formulation will emphasize more on $f(x)$, and hence, the solution is biased towards purely local models. (2) the "optimal" local steps derived in this paper becomes larger. Obviously, these two consequences are parallel to each other. We cannot say the second point is the reason of the first point. Instead of setting the expected local steps to be $1+L/\lambda$, one can also use other values which won't influence the final solution.

**Experiments**
The experimental results can only show the importance of variance reduction, which seems to be a minor contribution of the paper. Most theoretical claims are not validated empirically.

## Post-rebuttal
Thanks the authors for the clarifications! I appreciate it. However, some of my concerns are not addressed.
- The main concern I have is about the new insight on local update methods. Basically, the author obtain the insights based on a newly proposed algorithm (L2GD, let's call this algorithm B) and a new problem formulation (let's call this formulation B). However, they want to apply the insight from algorithm B and formulation B to algorithm A (original local update methods) and formulation A (original FL formulation). It is obvious that one cannot draw this conclusion because both the algorithm and the formulation are different.
- Second, as I stated in the original review, I don't think one can obtain the insights from the analyses in this paper. The author didn't directly answer my question and just said "they didn't expect people to interpret it in this way". But it is still unclear how to correctly understand their insights.

Based on the above two points, I strongly feel that their main insights about the effects of local updates should be further and carefully examined. The current version could be misleading. Besides, I also have the following minor concerns:

- The authors claim that [Yu et al. ICML 2019] didn't consider the heterogeneous setting. This is not true. Although [Yu et al. ICML2019] assumes that the gradient dissimilarity is uniformly bounded (which is widely used in literature), their setting is still non-iid setting. It's unfair to say that they only study the IID data setting. So the second motivation of this paper does not make sense to me. The authors oversell their contribution. More precisely, their contribution is not the first proof under data heterogeneous setting but should be the new proof without data similarity assumption.
- "non-local cousins" is unclear and hasn't been properly defined in the paper. For local SGD with mini-batch size ,  local steps and  clients, there are two non-local cousins: (a) SGD with mini-batch size ; and (b) SGD with mini-batch size . It seems that the authors misused these two algorithms. In the response, they agree that [Yu et al. ICML 2019] proves "with data dissimilarity assumption, local SGD can improve the communication complexity of classical SGD". Here, classical SGD refers to algorithm (a). In the updated paper, they cite two papers from Woodworth et al. to support their claim. However, the non-local methods in Woodworth et al. is algorithm (b). The authors should formally define which non-local algorithm they want to compare with.
- In the paper, the authors claim that they prove for the first time local methods can improve the communication complexity of the non-local cousins. However, this statement is overselling. The more precise version is that they prove that the variance-reduced version of local methods can improve the communication complexity of the vanilla non-local version algorithms.
- It seems that the authors want to claim a lot of contributions in this single paper and they didn't organize these contributions well. Hence, it causes difficulties for readers to understand their true novelty. I recommend the authors to rewrite the paper and carefully consider the paper structure. For example, if I understand correctly, the main contribution of this paper should be the insights on local updates. However, the authors didn't show any experiments on this insight in the main paper (they put them in the appendix). Instead, they just validate the effect of variance reduction in the main paper, which is just a minor point. Also, in the introduction, there is a long paragraph to introduce L2GD as one of the main contributions. However, as discussed in the responses, L2GD is not a new algorithm. The authors don't need to give it so much emphasize or should not claim it as one contribution.
- Also, in [1] EASGD does use multiple local steps. The authors should compare L2GD with EASGD, as they both are designed to minimize the new formulation.

## References

[1] Zhang et al. Deep learning with elastic averaging SGD. NeurIPS 2015.

[2] Wang et al. Overlap Local-SGD: An algorithmic approach to hide communication delay in distributed SGD. ICASSP 2020.

[3] Yu et al. On the linear speedup analysis of communication efficient momentum sgd for distributed non-convex optimization. ICML 2019.

---

> ### Author Response · Authors · 2020-11-24
> **Reply to "New insights about the role of local steps"**
>
> We thank the reviewer for the many questions and issues raised. Appreciated. Some are easily addressable and very minor, as we explain below. All remaining issues come from a fundamental misunderstanding by the reviewer of what our paper's key contributions are. We are happy to explain.
>
> We kindly ask the reviewer to read our response and reconsider their score, which we believe is unjustified.
>
> **New insights about the role of local steps: Issue 1**
>
> We start with this point because we believe that it the most crucial, and it would be a major issue if the mentioned criticism was valid. It is not. We insist that the claim "the role of local steps in gradient type methods is not to reduce communication complexity, as is generally believed. Indeed, there is no theoretical result supporting this claim in the key heterogeneous data regime"
> is correct.
>
> Note that the provided reference [3] {\bf does not consider the data heterogeneous setup.}; [3] considers either identical data or a bounded dissimilarity between local gradients (equation (3) therein) over the whole domain. Under such data similarity assumption, local SGD can outperform classical SGD (similar results have been provided in a convex case as well).
>
> However, and we do make this very clear in the paper, we are concerned with the much more difficult and important to FL setting: the data heterogeneous setting. That is, we do not use any kind of data similarity assumption whatsoever!
>
> Our paper is the first to show that (variants of) local SGD can outperform synchronous SGD in communication complexity *if no data similarity is assumed*. We show that this is the case when one aims to solve our new FL formulation (1), which happens to have a natural interpretation as a personalized FL formulation, with personalization level controlled by a single global parameter $\lambda$. In other words, we interpret the purpose of local steps: we claim they indeed do help in reducing communication complexity even in the heterogeneous data regime (and are the first to prove so), but in order for this to happen, we need to see them as methods for solving the personalized FL formulation (1) we propose. Moreover, as we show, (some variants of known and new) local methods can indeed be seen as methods for solving (1). We spend considerable real estate in the main paper to explain this on what we believe is the simplest of all local methods: local gradient descent (LGD). Indeed, LGD can be seen as SGD with importance sampling applied to (1) seen as a 2-sum problem. This alone, we believe, offers new conceptual insights to the FL community.
>
> We stress that in the practical FL applications, the often invoked bounded data dissimilarity assumption (between local gradients over the whole domain) does not hold: in such a case, local GD methods do not provide any benefit over their non-local cousins, yet they are still the most prominent FL optimizers.
>
> **New insights about the role of local steps: Issue 2**
>
> We did not intend this sentence to be interpreted the way you interpreted it. It was intended as an informal "intuitive" statement capturing some of the essence of our findings. It is not precise and was not intended to be. We will find a way to reformulate this sentence to avoid the kinds of confusion you are pointing out. However, notice this is a very minor point about a single sentence which is easily fixable.

---

> > ### Comment · AnonReviewer2 · 2020-11-24
> > **Could you please upload an updated version of the paper?**
> >
> > Thanks for the authors for the response! I'll reply to your responses in the near future. Besides, I notice that you mention that many concerns can be fixed easily and you will clarify some confusions in the final version of paper. But in order to better re-evaluate your paper, is that possible for you to upload an updated version during the rebuttal phase? Thanks!
> >
> > If you didn't intend to submit another version during this phase, then please just ignore this comment. You don't need to do it in a hurry since the deadline is approaching.

---

> > > ### Author Response · Authors · 2020-11-24
> > > **Thanks**
> > >
> > > We'll do what we can with the time we have, and we promise to do the rest, if anything remains to be done, later. Please consider our responses as well as they give the explanations. Note that the changes needed to address issues raised by all reviewers are minor. Most issues are rooted in a fundamental misunderstanding of our paper (and in the case of one reviewer, we believe, in failing to read the paper altogether).
> > >
> > > Thanks!

---

> ### Author Response · Authors · 2020-11-24
> **Reply to "New algorithm: Loopless LGD"**
>
> **New algorithm: Loopless LGD**
>
> **Issue 1: LGD is not new**
>
> We thank the reviewer for the reference [2]: good find! Indeed, the method is similar to the simplest of our methods (LGD) as it is a variant of local SGD which performs a step towards the averaging instead of the full averaging. We will properly cite the mentioned paper in the revised version, thanks!
>
> While the two methods (L2GD and the methods from [2]) are very similar in terms of the statement of the algorithm, this does not in any way decrease our key contributions as our main contributions are not algorithmic but methodological.
>
> We do design several methods in the paper, and LGD is the simplest of them all. We use it precisely because it offers a *very simple model example of a local method*, with just the right amount of structure, and without additional features our more advanced methods provide (see the appendix), enabling us to explain our main contributions clearly. We have developed and described much more advanced methods than LGD and the method in [2], but we start with LGD to make the narrative and explanations to be as simple and clear as possible. Now that we know LGD is similar to [2], we will mention this in the paper, but our narrative and contributions would not be diminished. After all, LGD is just SGD with importance sampling applied to (1) seen as a 2-sum problem, and hence is not a new method! Its interpretation and connection to FL is what matters.
>
> One can ask what is the advantage of our approach over the classical FL methods. The simple answer is: *we don't compete against them; we explain them!* Furthermore, most of the extensions we do in the appendix were not done in the classical FL setup; this is yet another contribution, albeit a minor one when compared to our key contribution: exhibiting the first link between local methods, personalization and communication efficiency. We believe this is of major import to the FL community.
>
> **Issue 2: Importance of random number of local steps**
>
> The fact that LGD uses a random number of local steps is not important. It is just *convenient* since it allows us to generate local methods as special cases of (existing and novel variants of) SGD applied to (1). It may be the case that a random number of local steps leads to slightly better bounds (e.g., in constants). However, we did not investigate this as this was not important for the purposes of our paper. However, the dependence on the rate of convergence (i.e., on $\varepsilon$) is certainly not affected by the use of random vs fixed number of local steps.
>
> Note that the method from [2] was analyzed as method for solving the classical FL objective ($\lambda=+\infty$).

---

> ### Author Response · Authors · 2020-11-24
> **Reply to "Convergence theory"**
>
> **Convergence theory**
>
> Indeed, we do ignore the second term in the convergence theorem of LGD. This is done *on purpose* because it is not important. Why? Because later on in the paper (Sec 5. and the appendix) we propose a way to remove the neighborhood (i.e., effect of $c$) completely (see footnote 5). This new variant of LGD is somewhat more complicated (it's a variance reduced SGD and not simple SGD), and we did not want to make the main body of the paper more complicated than needed to make our point. Indeed, LGD offers just enough structure for us to be able to make most of our main points clear. So, all our conclusions are valid, but some need to be seen as conclusions that apply to the more complicated methods we describe later on.
>
> We see that there is a mistake in the sentence above equation (9) (however, this is clarified by equation (9)): the $\epsilon$-neighborhood should be replaced by $\epsilon\|x^0-x(\lambda) \|^2 + \frac{2n\alpha \sigma^2}{\mu}$. Similarly, we will stress the target neighborhood size in the last paragraph of Sec 4.
>
> Lastly, we would like to stress that the statements of Theorem 4.1 and Corollary 4.3 are correct. Note that linear convergence of SGD *with a fixed stepsize* to a $O(stepsize)$ neighborhood of the optimum is a classical result, and not an issue with our analysis. See the paper of Gower et al (2019) whose theorem we are applying. If one wants to make the neighborhood small, e.g., $O(\epsilon)$, there are known techniques to achieve this. For instance, one may use a decreasing stepsize. We do not do this since this reduced the linear rate to a sublinear rate. Another approach is to choose a small stepsize. Indeed, if we set $\alpha = \frac{\epsilon \mu}{4n \sigma^2}$, then the constant error/neighborhood term in Eq (9) becomes bounded by  $\epsilon/2$, and one can achieve $\epsilon$ accuracy as long as $k=O(\frac{1}{\epsilon} \log \frac{1}{\epsilon})$. Again, this makes the method slower than linear. Instead, we employ control variates (see Section 5) to remove this term completely (notice that Corollary 5.2 does not have any such term).  However, we made a conscious choice to use LGD as the model method for explaining our main contributions, as we believe this will be more easily understood by more people.
>
> We will make this more clear in the camera ready version of the paper.

---

> ### Author Response · Authors · 2020-11-24
> **Reply to "New formulation" and "Experiments"**
>
> **New formulation**
>
> **Issue 1**
>
> Recall that $f(x)=\frac{1}{n}\sum_{i=1}^n f_i(x_i)$, where $x_1,\dots,x_n\in R^d$. Let $P(z) = \frac{1}{n}\sum_i f_i(z)$, where $z\in R^d$. Recall that for $x=(x_1,...,x_n) \in R^d \times \dots \times R^d$ we have $\bar{x}=\frac{1}{n}\sum_i x_i \in R^d$. We claim that  $$||\nabla P(\bar{x}(\lambda))||^2 = O(1/\lambda).$$
>
> This is the statement we referred to (and indeed, we did not include it in the paper). This means that as $\lambda\to \infty$, the gradient of $P$ evaluated at $\bar{x}(\lambda)$ goes to zero. Since $P$ is strongly convex, and since $x(\infty)$ is the unique solution of $\min P,$ we must have  $\bar{x}(\lambda) \to x(\infty)$.
>
>  Let us prove the claim. First, observe that $$||\nabla P(\bar{x}(\lambda))||^2 = ||\frac{1}{n}\sum_i \nabla f_i(\bar{x}(\lambda) ) ||^2 =  ||\frac{1}{n}\sum_i \nabla f_i(\bar{x}(\lambda) ) - \frac{1}{n}\sum_i \nabla f_i(x_i(\lambda) ) ||^2,$$ where the last identity is due to Theorem 3.2 which says that $ \frac{1}{n}\sum_i \nabla f_i(x_i(\lambda) )=0$. By applying Jensen's inequality and Lipschitz continuity of functions $f_i$, we get $$||\nabla P(\bar{x}(\lambda))||^2 \leq \frac{1}{n} \sum_i ||  \nabla f_i(\bar{x}(\lambda) )  - \nabla f_i(x_i(\lambda) ) ||^2 \leq \frac{L^2 }{n} \sum_i ||\bar{x}(\lambda) - x_i(\lambda)||^2 = 2 L^2 \psi(x(\lambda)).$$
>
> The claim now follows by applying Theorem 3.1 which says that $\psi(x(\lambda))=O(1/\lambda)$.  More can be established than this using similar arguments. For instance, a bound involving individual $x_i(\lambda)$ rather than $\bar{x}(\lambda)$, a $O(1/\lambda)$ bound on squared distance to $x(\infty)$ (the latter under an additional assumption).
>
> We will clarify in camera ready.
>
> **Issues 2 and 3**
>
> Indeed, the formulation itself was already considered earlier in the literature; we mention this already in footnote 1 (we will add reference [1]; thanks!). Having said that, i) we did not borrow this formulation from anywhere, we came to it through naturally our desire to understand the meaning of local steps in FL optimizers, ii) the formulation was never considered in the context of federated learning or local methods in particular, and it is this context and what we manage to explain what is important.
>
> The approach from [1] only takes a *single* local step in between of the communication rounds regardless of the value of the regularizer -- we have demonstrated that this is suboptimal and one should do more local work in between of the communication rounds.
>
> **Experiments**
>
> We *very strongly* disagree with the claim that "Most theoretical claims are not validated empirically." Only one of our experiments is designed to show the importance of variance reduction. In the appendix, we also study other aspects such as the effect of parameter $\lambda$ on the convergence speed and the effect of the communication frequency $p$ on the convergence speed (this is also demonstrated in Fig. 2). The reviewer missed this.
>
>
> [1] Zhang et al. Deep learning with elastic averaging SGD. NeurIPS 2015.
> [2] Wang et al. Overlap Local-SGD: An algorithmic approach to hide communication delay in distributed SGD. ICASSP 2020.
> [3] Yu et al. On the linear speedup analysis of communication efficient momentum sgd for distributed non-convex optimization. ICML 2019.

---

### Official Review · AnonReviewer3 · 2020-10-28
**Well-written paper; Concerns regarding novelty of the formulation and analysis, the role of penalty parameter, and comparison with related works.**

**Rating:** 4
**Confidence:** 4

**Review:**

Summary of the paper: The paper proposes a new formulation for the federated learning problem, in which each agent has its local model, and a penalty term is added to the objective function to control the deviation of these local models from their average. Next, the authors develop a randomized algorithm to tackle this problem and characterize its convergence under several assumptions, such as smoothness and strong convexity. They also discuss variants of their algorithm, which uses variance reduction techniques or considers users' partial participation.


The paper is well-written, and the goals, problem formulation, and contributions are all explained in detail. However, the reviewer has a number of concerns, which are listed below:



The first concern is regarding the novelty of the formulation or analysis. The idea of this paper's formulation, giving a copy to each agent and adding a regularizer to keep the copies close, has been discussed in the distributed literature, for instance, in ADMM. Moreover, it is not clear which part of the analysis is novel or challenging. It seems that the authors use an unbiased estimator to solve an optimization problem with a smooth and strongly convex objective function. This setting has been studied extensively in the literature, including applying variance reduction techniques.



Second, the regularizer parameter $\lambda$ seems to be at the heart of this framework. With $\lambda=\infty$, the problem reduces to the classic federated learning setting, and when $\lambda=0$, the formulation boils down to the case that each agent solves its own problem. In particular, for the latter, the authors claim that "such purely local models are rarely useful." However, this claim's reasoning is not clear; for instance, from a theoretical point of view, results such as Theorem 3.1 suggest that having $\lambda=0$ will lead to the minimum loss $f$. In other words, it is not clear how we should compare the different trained models from setting different values for $\lambda$, and which range of $\lambda$ leads to a good model with respect to that measure.



Third, as stated in the introduction, several methods have been recently proposed to address the heterogeneous case or achieve personalization in the federated learning problem. I wonder why the authors have only compared their methods against each other in experiments and have not included those methods for comparison.

---

> ### Author Response · Authors · 2020-11-24
> **First concern does not address our key contribution; and the second and third concerns are easily explained - not an issue with our work**
>
> First concern.
>
> Penalization is an old and well studied technique in optimization, and of course, we are not claiming novelty of this type.  Similar penalties were considered earlier in the literature in very different contexts; see footnote 1. Our main insight is that the formulation we propose is *new and particularly meaningful in the context federated learning (FL), resolving or at least giving important insights to several key issues in FL*. We spend a considerable space in the paper to explain this. Let us reiterate some of these points here:
>
> Despite a significant effort by the FL community, and despite the fact that this was the original and (still is!) motivation for their development, local methods (e.g., LGD) were never proven to reduce communication complexity when compared to their non-local counterparts in the heterogeneous data regime (see Woodworth et al. 2020 https://arxiv.org/pdf/2006.04735.pdf). There are also counterexamples which show this can't be done in general, even if one considers just convex quadratic functions. The belief that local steps are performed to reduced communication complexity is, we argue, one of the key confusions the FL community seems to suffer from, and our work is trying to remedy this situation. One can make several possible conclusions from this: i) local methods have better communication complexity, but we still do not know why as our analysis tools can't fully explain how well they work, ii) local methods are not actually beating non-local methods in terms of communication complexity, as it is universally asserted, and hence should be replaced by better performing methods, iii) local methods are good at solving a different problem of crucial importance to FL, but we do not know what problem it is.
>
> Our work is motivated by mounting evidence that ii) is true (in the heterogeneous data setting), which motivated us to think about iii) as a possible solution. Indeed, we manage to show that if one thinks of local methods as methods for solving our FL formulation (1) instead, then they become superior to non-local methods in communication complexity! This is the first time such superiority of local methods is shown to nonlocal methods in the heterogeneous data regime, which is the most important regime for FL! Hence, we believe our work is a major conceptual breakthrough in the field. See lines 108-116: communication complexity of LGD decreases to 0 as $\lambda \to 0$. So, if one aims for more personalization (corresponding to small $\lambda$), then local methods will have better and better complexity. In the extreme case when one requires absolute personalization ($\lambda=0$), each device is simply training a model from their own data only, and no communication is needed. So, this makes very good intuitive sense. Note that, for example, LGD (one of the most basic variants of FedAvg) can be interpreted as a SGD with importance sampling applied to (1) seen as a 2-sum problem. Yes, we did not have to come up with a new analysis for SGD in this case; we clearly explain this in the paper. So, our novelty here is not in the analysis, which, as you say and as we say in the paper, simply follows from Gower et al (2019). The novelty is the insights that SGD applied to (1) in the way we do it generates LGD! So, the mystery of the utility of local steps evaporates: they are there to put more emphasis on $f$ instead of the penalty. However, more emphasis of this type is desired precisely when $\lambda$ is small, i.e., when we require more personalization. So, the key novelty of our paper is that we connect personalization, communication complexity and local methods together, in an insightful manner. Our several new local methods are a secondary contribution.
>
> Second concern.
>
> We study the effect of how $\lambda$ influences the convergence rate and the optimal # of local steps. We do not study which choice of $\lambda$ is better from a generalization perspective and hence we can't give a theoretical prescription to practitioners of this type.
>
> What we mean by saying that "such purely models are rarely useful" is this: in practice, devices do not have enough data to be able to train models using their own data only, which is why we need to resort to distributed methods such as FedAvg or LGD.  If they had enough data, $\lambda=0$ would be a perfectly fine choice, and there would be no need to do any communication. So, $\lambda=0$ would be optimal and FL in such a data-rich regime would simply reduce to $n$ independent training problems performed by the devices independently. Yes, $\lambda =0$ leads to the smallest training loss. However, this does not mean we would get the smallest testing loss.
>
> Third concern.
>
> The reason is simple: the other methods solve a different optimization problem and hence are not comparable. Note that none of these methods outperform their non-local cousins in terms of the comm. complexity in the heterog. data setting.

---

> > ### Comment · AnonReviewer3 · 2020-11-25
> > **Response to atuhors**
> >
> > Thank you for your response.
> >
> > Regarding $\lambda$:
> > Yes, I understand your point that $\lambda=0$ is not good in practice since each user does not have enough data to train its own model. At the same time, if we just look at the theory (for instance, Theorem 3.1), $\lambda=0$ seems to be the best choice as it gives the ultimate personalization. Hence, there appears to be a trade-off between choosing small $\lambda$ and aiming for more personalized models and larger $\lambda$ and somehow taking more advantage of all users' data over the distributed network. Hence, $\lambda$ is the parameter that governs an underlying and important trade-off here. However, your result does not offer guidance on choosing $\lambda$, which seems to be a critical missing piece in your proposed framework.
> >
> > Regarding the third concern (comparison with other works):
> > I understand that your formulation is different, but I still think you can compare your method with other works. Your paper and those related works all try to come up with different ideas and methods for adding personalization to federated learning. Each of these works ends up proposing a model to be implemented on each device. For instance, your paper suggests implementing $x_i$ for device $i$. Other paper suggests a different personalized model. But still, these methods can be compared, in terms of test accuracy (with an example with heterogeneous data distributions), or in terms of communication efficiency, etc. In other words, while I understand the formulations are different, the final outcomes can be compared to see which one obtains better personalization, which one is more efficient in terms of communication, etc.

---

> > > ### Author Response · Authors · 2020-11-25
> > > **Reply to: $\lambda$ and third concern**
> > >
> > > About $\lambda$:
> > >
> > > Yes, our theory does not provide guidance on how to choose $\lambda$. We focus in this paper on many other aspects of our setup, but not on this one. The reason for this is not a simple oversight. In order to give guidelines on what $\lambda$ to recommend, we would need to i) tackle the problem of generalization (as that is the ultimate arbiter here), ii) do extensive systems-level testing. These are nontrivial matters that require a separate paper. Our paper is not about generalization and is hence we are not able (i.e., we have no theory fo this) to recommend some $\lambda$ other than another from this applied perspective. At present our recommendation is this: i) practitioner (somehow) decides on level of personalization and chooses $\lambda$, ii) we then recommend training is done b solving our personalized FL formulation (1).
> > >
> > > Third concern:
> > >
> > > Yes, what you say is absolutely right; we agree with this - this kind of a comparison would make perfect sense, and is something we were actually thinking of when writing the paper. However, we made a conscious choice not to further broaden the scope of this paper as we already had an abundance of results which stand on their own. So, we decided to keep this paper about optimization and not generalization and decided to keep testing against other personalization benchmarks for a future project. In fact, we have been working for some time now on exactly this problem. We wish to compare many FL methods from a personalization perspective, including ours. This is something that in our mind requires a dedicated paper.
> > >
> > > We can think of many things that could be added to our work. The beauty of research is that there are always frontiers ahead, always something new to discover. But one ultimately needs to decide where one paper ends and another starts. We made the line here: we purposefully decided to not go nonocnvex in our paper, and we also decided not to go into benchmarking and generalization comparison in this work. We believe that the results we have obtained stand on their own. And we believe that our work will inspire others as well to explore these ideas further.

---

### Official Review · AnonReviewer4 · 2020-10-29
**Authors propose a model personalization method for federated learning with a new optimization formulation which provides an explicit trade-off between the global and local models. In addition, the authors develop several efficient variants of SGD for solving the new formulation and prove communication complexity guarantees.**

**Rating:** 4
**Confidence:** 4

**Review:**

***Strong

Personalization is a hardcore problem in FL. The authors target an important problem.

The theory analysis seems correct, but the bound seems not tight enough.

***Weakness

Recently, there are many personalized methods proposed for FL. In the I.I.D. setting, local SGD training (FedAvg) can obtain similar accuracy as centralized training with a theory guarantee.  But in the non-I.I.D. setting, I am curious to know what’s the ultimate goal of optimization. Can the proposed personalized methods obtain accuracy comparable to centralized training? If we do not compare with the centralized accuracy, how can we know the optimized personalized model can obtain sufficient accuracy for practical applications.

The experimental results are weak. The authors only provide results on the LR model for toy datasets (LibSVM). Without non-convex experiments, it is hard to believe the proposed method works in practice given that DNN-based models dominate nearly all ML tasks.

The code style and readability are poor, which discourages the popularity of the proposed method.

Although the authors mentioned some contributions of the proposed method, I still cannot get what’s the advantages of the new formulation over conventional Federated optimization? What is its limitation? What’s the cost to use this method? When should we choose this algorithm in practice? In which degree of non-IIDness? Playing with optimization analysis tricks won’t solve the personalized challenge of federated learning in practice.

---

> ### Author Response · Authors · 2020-11-22
> **This review is superficial and utterly confused.**
>
> This review does not seem to address any substance actually contained in our paper. Instead, it offers general philosophical thoughts related to the general theme of our paper. It is not possible to meaningfully respond to a review of this type. These kids of reviews are not helpful, and actually inflict quite a bit of harm on the community.
>
> - "The bound seems not tight." What bound? We have many. We believe we offer several efficient algorithms, starting with simple ones which are easier to understand (for pedagogical/clarity reasons), and gradually adding features and enhancements (e.g., adding control variates, partial participation and so on). Your comment is generic and does not address our work. It could have been made without actually reading our paper at all. This comment should be ignored by the AC.
>
> - First paragraph in weaknesses: you ask some questions but our paper is not about this. Our contributions are clearly stated and you do not refer to them. You do not seem to have a genuine interest in what we actually accomplished.
>
> - Experiments: The results are not weak. Our theory is for convex problems, and the methods are fine-tuned for convex problems. We test the theoretical predictions with carefully designed experiments and observe that our theory predicts what happens in experiments very well. This is the ideal scenario of any scientific work containing theory. The experiments are strong. Testing our methods in the nonconvex regime does not make much sense; we would need to first develop the associated theory, and this is beyond the scope of the current paper. Not all FL tasks are deep learning tasks.
>
> - Code: You criticize the readability of our code, but offer no concrete evidence of what is wrong. Again, this is a generic comment that could have been made about any paper containing a method. This kind of a comment is not helpful. If you found concrete issues, list them. We are happy to correct and improve our paper, but we can't do this if issues are not pointed out to us.
>
> -  Last paragraph: These are again comments divorced from the actual contents and contributions of our paper. It is very clear to us that this reviewer simply failed to understand our work and contributions. Or perhaps the reviewer even did not read the work properly. In any case, it is not possible to respond meaningfully to a review that fails to address the evaluated paper to such a degree as this review manages to do so.

---

> > ### Comment · AnonReviewer4 · 2020-11-24
> > **My comments are reasonable**
> >
> > The authors solely criticize my comments but do not answer questions in depth. Let me make my comments more concrete.
> >
> > The fundamental question in Paragraph 1 is not superficial. Please answer it in depth. If the accuracy loss is high in non-IID setting, why don't we directly centralized encrypted data to a secure data center and delete data after completing the training? We need to solve the hard core problem rather than say you have many contributions.  From other reviewers' comments, I believe my judgement (See R3).
> >
> > I don't agree with the authors' comments in deep learning. If the proposed convex optimization method can provide intuition for DNN-based algorithms, please demonstrate in a small CNN/RNN. LibSVM is a toy dataset nowadays, thus the experiments cannot convince me at all, even in theory.
> >
> > Code: It seems the authors update the code recently. This time I think the code is clear to me. But I still have some suggestions:
> > 1) make the variable readable using a longer name.
> > 2) warp the key contribution/algorithm/feature with a small function/class, so others can directly reuse it without the need to do code extraction.
> > 3) Add more comments and make a good README.md file, telling readers what's the functionality of each class and file.
> >
> > My comments in the last paragraph means a better presentation to understand your work, although I use many questions to express my concern. Please make the applicability of the proposed method more clear to readers in revision.

---

> > > ### Author Response · Authors · 2020-11-24
> > > **No, your comments are not reasonable**
> > >
> > > Yes, we criticize the review since we can recognize a bad review when we see it. Your is very bad. Uninformed, vague, not addressing anything that is actually contained in our paper, and yet critical. You create a strawman and attack it. We believe it is our duty to call this out.
> > >
> > > The question mentioned in Paragraph 1 does not address our paper. It addresses and questions all of non-iid federated learning in its entirety. You are attacking a field here. You are asking questions which are not related to our paper. We do not deal with generalization, we do not compare to any centralized training mechanism. We solve an entirely different problem from the one you want our paper to solve. Please note you can't criticize paper on topic A (what we do) because it did not solve topic B (what you suggest we should investigate instead; which we do not).
> > >
> > > We are open to constructive criticism that addresses specific issues in our paper. But no one can be expected to reply to vague and irrelevant comments. It's not possible.
> > >
> > > Rev 3: As a reviewer, do not just believe a reviewer. Read the paper and make up your own mind. As we argue in the reply to Rev 3, the criticism is based on misunderstanding, and not because there is an issue with our work. Please read our reply.
> > >
> > > Re experiments: You say "If the proposed convex optimization method can provide intuition for DNN-based algorithms..." The premise is false, and hence the argument is false. We do not claim our method (here is another indicator you did not read or understand our paper - our paper is not primarily about a method) provides intuition for DNN-based algorithms. We do something that until now was not understood even in the convex case, and the correct scientific approach is to work on that first. Only then can one meaningfully attack the nonconvex problem. In the nonconvex regime, we would need to consider different methods with very different theory and parameter settings. The nonconvex case is beyond the scope of this work. We do not wish to run a heuristic, we want to develop some guiding theory first.
> > >
> > > Code: Thanks for the good suggestions.
> > >
> > > Last paragraphs: Many of these questions are answered in the paper. Please read it. And many are irrelevant to our work. So, it is hard to respond to this.
> > >
> > > The correct way to write a review is to be very detailed and specific about what is being criticized and why. Your review is on the opposite side of the spectrum, and as such ranks among the worst / least helpful / most confusing reviews we have ever read.

---

### Decision · Program_Chairs · 2021-01-07
**Final Decision**

**Decision:**

Reject

**Comment:**

The paper studies an elegant formulation of personalized federated learning, which balances between a global model and locally trained models. It then analyzes algorithm variants inspired by local update SGD in this setting. The problem formulation using the explicit trade-off between model differences and global objective was received positively, as mentioned by R1 and R2. After a productive discussion including the authors and reviewers, unfortunately consensus remained that the paper remains below the bar in the current form. The contributions are not presented clearly enough in context, the set of competing algorithms (including e.g. EA-SGD, ADMM, SVRG/Scaffold for the heterogeneous setting, and others) needs to be clarified in particular for the modified formulation compared to traditional FL, since objectives are different. Some smaller concerns also remained on the applicability to more general non-convex settings in practice. We hope the feedback helps to strengthen the paper for a future occasion.